# Accurate atomic resolution XFEL structures of a metalloenzyme reveal key insights into its catalytic mechanism*

Samuel L. Rose [1,6,7], Svetlana Antonyuk[1,7], Felix M. Ferroni [2], Hiroshi Sugimoto [3], Keitaro Yamashita [4], Kunio Hirata [3], Hideo Ago [3], Go Ueno [3], Hironori Murakami[3], Robert. R. Eady[1], Takehiko Tosha [5] ✉, Masaki Yamamoto [3] ✉ & S. Samar Hasnain [1] ✉

Metalloproteins represent a major fraction of the protein kingdom and often exploit the redox chemistry of transition metals to drive key biological events involving proton and electron transfer. Copper is one of the most widely used transition metals whose redox properties are utilised in both electron transfer and catalysis of chemical substrates. Copper nitrite reductases (CuNiRs) utilise two types of copper centres and have become a model system for studying complex biological events that underpin the reaction mechanisms of redox enzymes, including proton-coupled electron transfer and substrate gating. We utilised the higher X-ray energy (13 keV) available at the SACLA X-ray Free Electron Laser (XFEL) and SHELXL refinement to obtain accurate atomic resolution structures of CuNiRs at ~1 Å from three organisms – in the oxidised (low and high pH), reduced and substrate-bound states. A consistent picture now emerges with the observation of a pentacoordinated oxidised catalytic type-2 Cu (T2Cu$^{2+}$) centre in all cases. A tetracoordinated reduced T2Cu$^+$ site with a single solvent ligand has also been captured, giving structural support to the random-sequential scheme with ordered pathway being dominant.

Copper is one of the most commonly occurring redox-active metals in enzymes and plays a significant role in chemical biology. Its diverse redox-active chemical reactivity enables it to participate in various complex biological processes. Its ability to change reversibly between the Cu(II)/Cu(I) redox state during catalysis and electron transfer processes, while remaining coordinated to the amino acid ligands, gives it versatility[1]. These chemical properties are harnessed in a wide variety of oxidoreductases, including cytochrome $c$ oxidase[2], superoxide dismutase[3], tyrosinase[4], oxygenases[5] and nitrogen oxide (NO$_x$) reductases[6].

Copper nitrite reductases (CuNiRs) catalyse the reduction of nitrite (NO$_2^-$) to nitric oxide (NO) and are a key enzyme in the denitrification step of the nitrogen cycle[6,7]. The redox reaction involves two protons, one electron and the chemical substrate nitrite: NO$_2^-$ + e$^-$ + 2H$^+$ ⇆ NO + H$_2$O. Two distinct classes of nitrite reductases are found in denitrifying organisms: a cytochrome $cd_1$ type, encoded by $nirS$ and

[1]Molecular Biophysics Group, Life Sciences Building, Institute of Systems, Molecular and Integrative Biology, Faculty of Health and Life Sciences, University of Liverpool, Liverpool, UK. [2]Departamento de Física, Facultad de Bioquímica y Ciencias Biológicas, Universidad Nacional del Litoral (UNL). CONICET, Santa Fe, Argentina. [3]RIKEN SPring-8 Center, Hyogo, Japan. [4]Research Center for Advanced Science and Technology, The University of Tokyo, Tokyo, Japan. [5]Graduate School of Life Science, University of Hyogo, Hyogo, Japan. [6]Present address: European Synchrotron Radiation Facility, Grenoble, France. [7]These authors contributed equally: Samuel L. Rose, Svetlana Antonyuk. *Manuscript is dedicated to George Sheldrick (1942-2025) and represents the first application of SHELXL to atomic resolution macromolecular structures obtained using XFEL. ✉e-mail: ttosha@sci.u-hyogo.ac.jp; yamamoto@riken.jp; s.s.hasnain@liverpool.ac.uk

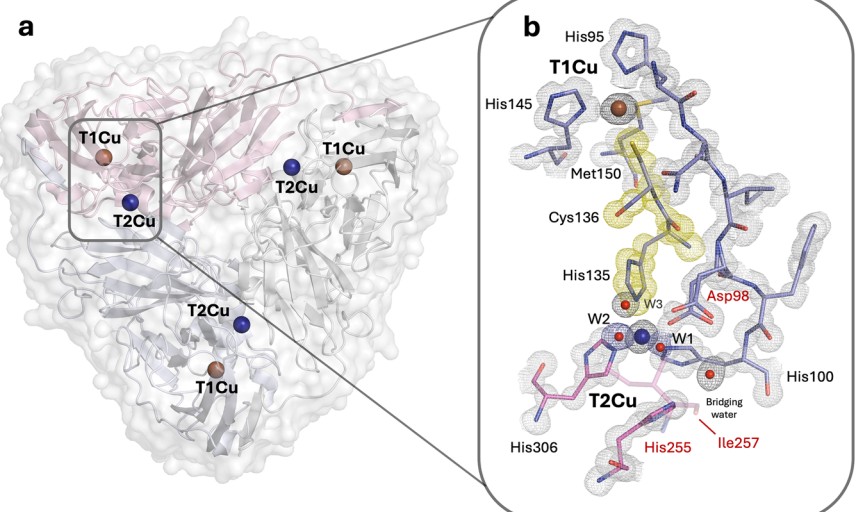

**Fig. 1 | Overall structure of trimeric CuNiR. a** Each of the monomers of the homotrimer possesses a T1Cu-T2Cu catalytic core where the T1Cu and T2Cu are bridged by the neighbouring residues Cys136 and His135 which are ligands of T1Cu and T2Cu, respectively, and represent the gated internal electron transfer route. **b** Expanded view of the T1Cu and T2Cu region.

the more widely distributed copper-containing enzymes encoded by *nirK*[6,7]. The Cu-containing nitrite reductases have been extensively studied and shown to be a trimer with each monomer having two domains with a characteristic β-sandwich motif[8,9]. Each monomer contains a type-1 Cu (T1Cu) centre close to the protein surface and a type-2 Cu (T2Cu) site at the interface between two monomers (Fig. 1). In the oxidized enzyme, the T1Cu shows strong absorbance bands in the visible region and variation in their intensity results in the colour varying from blue to green depending on the organism[10]. The T1Cu centre accepts electrons from either a cytochrome *c* or a cupredoxin and are transferred to the T2Cu centre, where the chemical reaction for the conversion of $NO_2^-$ to NO occurs[11]. Two invariant residues, catalytic Aspartic acid residue ($Asp_{CAT}$) and catalytic Histidine residue ($His_{CAT}$), located in the active site pocket, have a proposed role in proton delivery during catalysis, and an $Ile_{CAT}$ residue provides steric modulation of ligand binding to the T2Cu[12,13]. $Asp_{CAT}$ is mobile and has been observed in three different positions termed "gatekeeper" (pointed away from the T2Cu), "proximal" (orientated towards the T2Cu and H-bonded to a liganded $H_2O$) and distorted proximal, where the carboxyl $O^{δ2}$ atom is tilted closer towards a T2Cu ligand[14–16]. Nitrite binds to the T2Cu by displacement of a $H_2O$ ligand, forming an adduct that shows some variation in symmetry and "top hat" and "side on" conformations[14–17]. The two Cu centres, separated by 12.5 Å, are directly linked *via* a unique Cys-His bridge formed by neighbouring residues that are ligands to the two Cu centres. Both copper centres cycle reversibly between the Cu(II)/Cu(I) redox state during catalysis, a feature in common with redox centres of many key metalloenzymes such as cytochrome *c* oxidase[2], hydrogenases[18,19], nitrogenases[20,21] and nitrite reductases[7]. In these systems, as with CuNiRs, catalysis involves the controlled delivery of electrons and protons to the active site, where chemical substrates are utilised. The amenability of CuNiRs for biophysical study and highly diffracting crystals capable of providing high-resolution crystallographic structures have made them an attractive model system for studying these processes with a view of establishing general principles that underpin catalytic redox reactions in these systems.

In CuNiRs, it has been demonstrated that inter-Cu electron transfer (ET) is proton-coupled and during catalysis it is gated by nitrite binding to the catalytic T2Cu centre[22–24]. Recent high-resolution MSOX (multiple structures from one crystal) studies, combined with on-line single crystal spectroscopy during substrate-free and substrate-bound states, have provided evidence for the electronic route of proton-coupled electron transfer (PCET) from T1Cu to T2Cu ($S(σ)_{Cys}$ T1Cu$^+$→ T2Cu$^{2+}$)[25–28].

In general, redox chemical reactions during biological catalysis are often accompanied by only subtle changes (< 0.1 Å in bond length) in the coordination sphere, which can only be determined at very high resolutions approaching atomic resolution -1.2 Å. Such resolutions have been achieved using synchrotron radiation-based crystallography for CuNiRs[14,16,25]. However, the X-rays from synchrotron sources used to obtain high-resolution structures themselves introduce redox activation of these enzymes in the crystals during a typical crystallographic data collection. The photoreduction of the T1Cu centre in crystals results in the bleaching of the T1Cu optical spectrum, while in the nitrite (substrate) bound crystals it results in the conversion of nitrite to NO alongside the bleaching of the optical spectrum[22,26–28]. This serious limitation of unintended redox changes has been overcome by the use of X-rays from Free Electron Lasers (XFEL), which has allowed single-shot experiments using femtosecond pulses of X-rays. This enables time-frozen (-10 fs) structures to be obtained that are essentially free from radiation-induced chemistry (FRIC structures). Both XFEL-based crystallographic approaches, namely serial femtosecond crystallography (SFX) and serial femtosecond rotational crystallography of a single crystal (SF-ROX), have been used for CuNiRs. Single-shot XFEL experiments have provided much useful information on pure resting state and substrate-bound states, but the resolution has remained limited -the highest resolution for an SF-ROX structure was 1.3 Å[16] and for SFX was 1.6 Å[17]. The recent availability of higher X-ray energy (<13 keV) at the SACLA XFEL has enabled structure determination to atomic resolution <1.2 Å[29] where refinement of anisotropic displacement parameters (ADPs) and estimation of standard uncertainties (esds) through unrestrained refinement can be performed in SHELXL[30].

Here, atomic resolutions achieved using higher energy (13 keV) X-rays from an XFEL (Supplementary Table 1) is combined with full SHELXL refinement (including one cycle of unrestrained refinement) for refinement of ADPs and to estimate esds of bond lengths and angles around metal centre sites in a metalloprotein system (CuNiR). This has been done for several important functionally relevant states (low and high pH oxidised, substrate-bound, and chemically reduced states). It has been possible to collect data for many states in a limited allocated time using the recently implemented automatic data collection pipeline for our SF-ROX data collection at SACLA. These

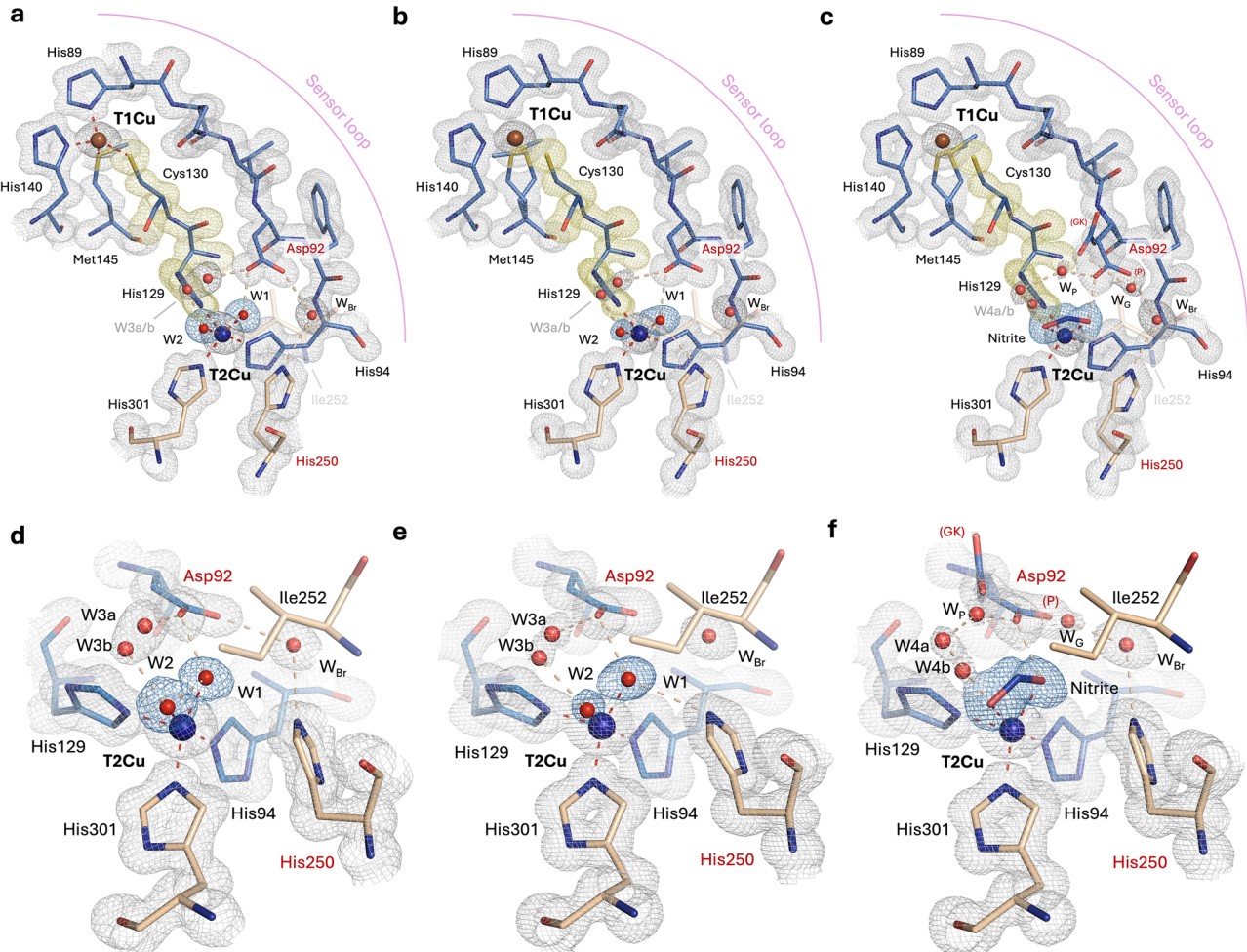

**Fig. 2 | Atomic resolution XFEL structures of the oxidised Cu(II) sites of *Bra-dyrhizobium* CuNiRs (*Br*NiRs). a** The atomic resolution (1.15 Å) XFEL structure of as-isolated *Br*NiR at pH 5.5, showing the T1Cu and T2Cu sites. **b** The atomic resolution (1.00 Å) XFEL structure of as-isolated *Br*NiR at pH 7.3, showing the T1Cu and T2Cu sites. **c** The atomic resolution (1.02 Å) XFEL structure of nitrite-bound *Br*NiR, showing the T1Cu and T2Cu sites. **d** The atomic resolution T2Cu active site of as-isolated *Br*NiR at pH 5.5. **e** The T2Cu active site of as-isolated *Br*NiR at pH 7.3. **f** The T2Cu active site of nitrite-bound *Br*NiR at pH 5.5. The four-residue sensor loop that connects the His ligands of the T1Cu and T2Cu sites and contains Asp$_{CAT}$ has a proposed sensor role in communicating the binding of nitrite to the T2Cu to the T1Cu site and is shown in (**a**)–(**c**). Residues from chain A are coloured in blue and chain B residues are coloured in gold. Residues are labelled in black and catalytic Asp and His residues are labelled in red. Waters are coloured in red. P proximal Asp$_{CAT}$ conformation. GK gatekeeper Asp$_{CAT}$ conformation, W$_{Br}$ bridging water, W$_G$ gatekeeper water, W$_P$ proximal water. $2F_o$-$F_c$ density map is contoured to 1 σ and coloured in grey (blue for T2Cu ligands and yellow for Cys-His bridge).

structures offer unprecedented accuracy and reveal several features of chemical importance in CuNiRs, namely the consistent presence of two solvent molecules bound to the oxidised T2Cu(II) in a five-coordinated site for both the less catalytically efficient bluish-green CuNiRs from *Bradyrhizobium* species (*Bradyrhizobium sp. ORS 375*[31] or *Bradyrhizobium japonicum*[32] - *Br*NiR) in their resting state, and also the prototypic green CuNiR from *Achromobacter cycloclastes* (*Ac*NiR)[14]. The mode of substrate binding for both also indicates a consistent five-coordinated T2Cu(II) site with structural and spectroscopic evidence for the strong preference of substrate binding to the oxidised T2Cu(II) before inter-Cu ET occurs. The coordination geometry is altered upon reduction to T2Cu(I). These observations collectively support that the binding-before-reduction[33] branch of the proposed "random sequential mechanism"[34] predominates during turnover.

## Results
### Accurate atomic resolution XFEL structures of *Bradyrhizobium* CuNiRs
Our updated (automated high-energy) SF-ROX data collection strategy for atomic resolution XFEL structure determination was applied to obtain cryogenic (100 K) XFEL structures of *Bradyrhizobium* CuNiRs

(*Br*NiRs) from either *Bradyrhizobium sp. ORS 375* or *Bradyrhizobium japonicum* at resolutions of 1.00 Å to 1.15 Å - for as-isolated (fully-oxidised) states at pH 5.5 and pH 7.3, a nitrite-bound and dithionite-reduced state, respectively (Supplementary Table 1,2). With the extension of the resolution limits to ≤1.15 Å, unrestrained SHELXL refinement[30] could now also be performed for these states, providing highly accurate bond distance and angle information. The current structures, together with one cycle of unrestrained refinement for estimation of standard uncertainties (esds) [provided in brackets], represent an example of redox enzyme systems, where XFEL structures, free from radiation induced chemistry, are obtained for several key functional states at such high resolutions.

### As-isolated oxidised *Br*NiR (pH 5.5) at 1.15 Å resolution
A common observation for these *Bradyrhizobium* CuNiRs has been the presence of two water molecules coordinated to T2Cu in the resting state active site, representing a pentacoordinated architecture[16,26,28]. The atomic resolution XFEL structure of as-isolated oxidised *Br*NiR at pH 5.5, determined at 1.15 Å resolution, was obtained and again revealed this coordination with two full occupancy water molecules (W1 and W2) bound to oxidised T2Cu(II) (Fig. 2a, d) - an F*o*-F*c* OMIT map is also shown for clarification

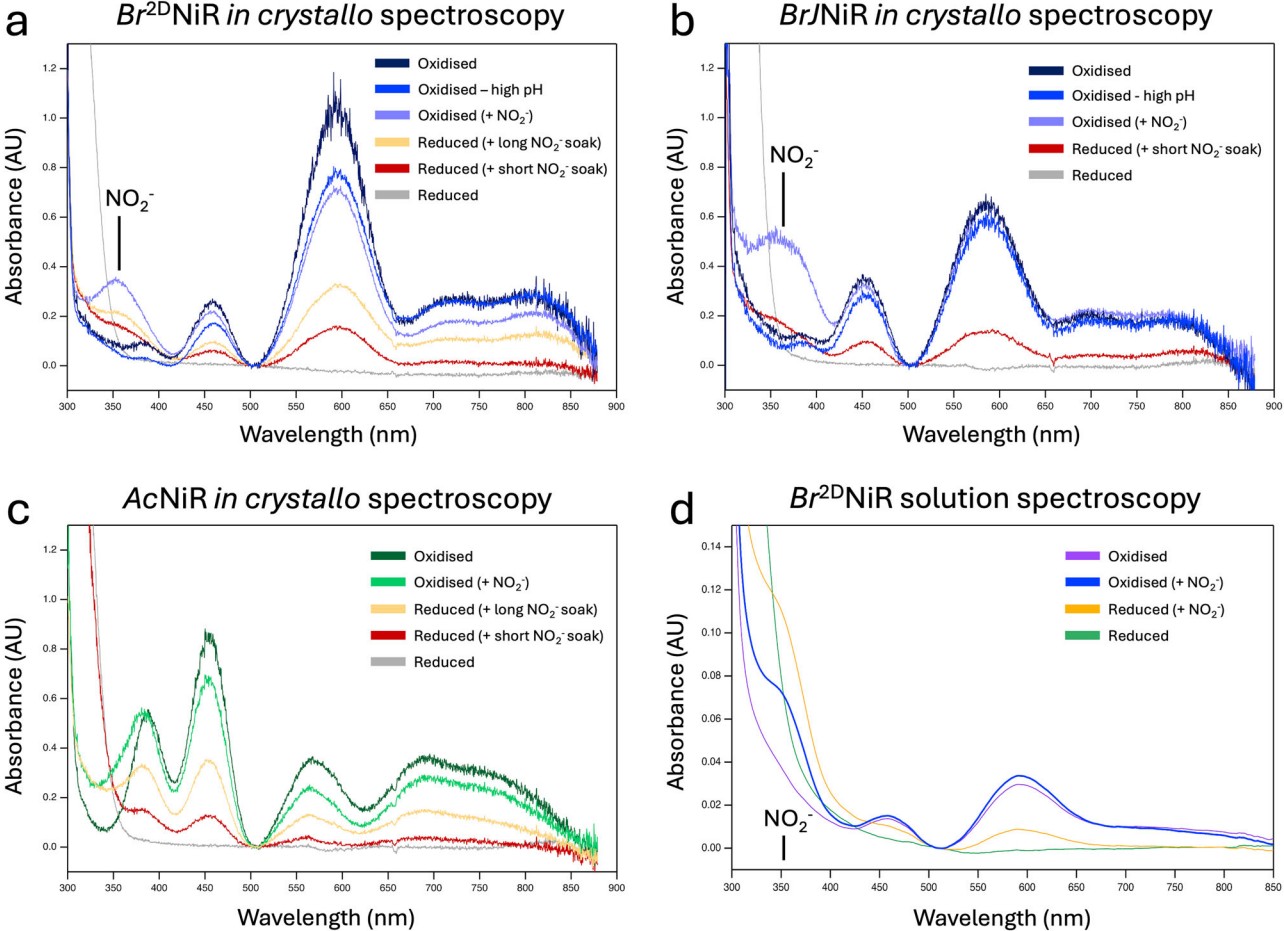

**Fig. 3 | Optical spectra of single crystal and solution samples of CuNiRs.** Single crystal spectra for **a** $Br^{2D}$NiR at pH 5.5 or pH 7.3 (high pH), **b** $BrJ$NiR at pH 5.5 or pH 7.3 (high pH), and **c** $Ac$NiR at pH 4.8. In each case, several spectra are presented. The starting spectrum are those of the oxidised enzymes, clearly showing that T1Cu is oxidised, to which nitrite was subsequently added. Nitrite-soaked crystals show a clear LMCT band at 350 nm following the addition of nitrite. In the case of the green $Ac$NiR it overlaps with the 380 nm peak of the oxidised enzyme. Grey curves in each of the panels were obtained from dithionite-reduced crystals of the oxidised enzyme. The loss of optical features confirms the T1Cu(I) oxidation state. Nitrite was added to these reduced crystals to probe the binding of nitrite to the reduced T2Cu site and if enzyme turnover results in inter-Cu electron transfer leading to partial recovery of the Cu(II) T1Cu site spectrum. A progressive partial recovery (~20%) was observed at longer soak times. The short soak time is the same as that required to obtain fully oxidised nitrite-bound enzyme. The longer soak was for a significantly longer duration. **d** Equivalent solution spectra of a *Bradyrhizobium* CuNiR (here $Br^{2D}$NiR) where nitrite was added to the enzyme in solution at pH 6.5.

(Supplementary Fig. 1a) However, in this structure the coordination and bond distances are slightly altered compared to the equivalent cryogenic XFEL structure at 1.3 Å resolution[16] (coordination distances are given in Supplementary Table 3 and Supplementary Fig. 2a.). In the current structure, W2 is coordinated to the T2Cu(II) at 2.08 (2) Å, similar to the lower resolution (1.3 Å) SF-ROX structure but W1 is at a considerably shorter coordination distance of 1.83 (2) Å compared to 1.94 Å seen previously. The two waters are also separated by a shorter distance of 2.23 Å. The shortened W1-Cu(II) distance of 1.83 (2) Å may indicate the presence of a hydroxide (OH⁻) ion ligated to T2Cu(II), which also hydrogen bonds to the proximal position of the catalytic Aspartic residue (Asp92; $Asp_{CAT}$), observed in this structure. $Asp_{CAT}$ also forms hydrogen bonds to an additional water (W3) in the active site that appears to be mobile. These observations are qualitatively similar to the lower resolution XFEL structure, albeit W3 being mobile as opposed to being well-defined. The catalytic Histidine residue (His250; $His_{CAT}$) bridges $Asp_{CAT}$ through a conserved bridging water ($W_{Br}$) and its imidazole ring is rotated towards Glu274, with its $N^{\delta1}$ atom hydrogen bonding with the carbonyl oxygen of Glu274 at 2.70 Å (Supplementary Fig. 3a). This is consistent with the previous SF-ROX structure at 1.3 Å resolution[16].

The optical *in crystallo* spectrum of one of the crystals prior to XFEL collection confirmed the T1Cu site to be fully oxidised (Fig. 3a). The atomic resolution XFEL structure of the oxidised crystal showed a typical distorted tetrahedral geometry for T1Cu(II), with slight elongation of the Met-T1Cu(II) coordination and shortening of Cys-T1Cu(II), more akin to a blue CuNiR compared to green (Fig. 2a and Supplementary Fig. 4a). The T1Cu(II) site shows no difference to the previous lower resolution-SF-ROX structure (coordination distances are given in Supplementary Table 3 and Supplementary Fig. 4a).

### As-isolated oxidised $Br$NiR (pH 7.3) at 1.00 Å resolution

The catalytic activity of CuNiRs, including the tethered CuNiRs with additional tethered redox domains, shows a bell-shaped pH profile with an activity maximum around pH 5.5-6.0[23,32,35-37]. The bell-shaped profile is attributed to the involvement of $Asp_{CAT}$ and $His_{CAT}$ in proton delivery. Here, in addition to pH 5.5, we have also undertaken XFEL structure determination of the as-isolated oxidised $Br$NiR at high pH (pH 7.3) to 1.00 Å resolution (Supplementary Table 1,2), This represents the only pH-dependent damage-free XFEL structure of any CuNiR and shows that the general T2Cu(II) architecture is markedly different compared to the lower pH structure.

The 1.00 Å resolution of this XFEL structure, together with SHELXL refinement, again suggests the potential presence of a T2Cu(II) OH$^-$ ligand in this state instead of a water molecule. The T2Cu site shows a highly ordered ligand (H$_2$O/OH$^-$; W1) coordinated in a perfect tetrahedral geometry relative to the histidine plane and a half-occupancy water (W2) in a position similar to W2 in the low pH structure (Fig. 2b, e) - a F$o$-F$c$ OMIT map is also shown for clarification (Supplementary Fig. 1b). The H$_2$O/OH$^-$ ligand (W1) is tightly bonded to W2 by 2.16 Å with a relatively short W1-Cu coordination distance of 1.90 (1) Å. W2 has a longer T2Cu(II) coordination distance of 2.20 (2) Å (Supplementary Fig. 2b and Supplementary 3). Unusually, the H$_2$O/OH$^-$ (W1) Cu ligand also appears to bridge between the two catalytic residues – Asp$_{CAT}$ and His$_{CAT}$, as opposed to the bridging water (W$_{Br}$) normally observed, with this bridging mode not observed before for any CuNiR. Favourable hydrogen bonds are formed with the carboxyl oxygen (O$^{\delta 2}$) atom of Asp$_{CAT}$ (2.55 Å) and the N$^{\epsilon 2}$ atom of His$_{CAT}$ (2.87 Å). In addition, a dual-occupancy partial water (W3) is also seen within the T2Cu active site pocket, providing some linkage between W2 and Asp$_{CAT}$. His$_{CAT}$ is also favourably rotated towards Glu274 with a hydrogen bonding distance of 2.71 Å (Supplementary Fig. 3b).

The coordination distances at the oxidised T1Cu(II) site are consistent with a fully oxidised species and were confirmed by an optical *in crystallo* spectrum taken on a crystal prior to XFEL collection (Fig. 3b.). The T1Cu(II) Met145 ligand shows two positions, whilst maintaining coordination, with the distances of 2.55 (1) Å (69 %) and 2.72 (1) Å (31 %), (Fig. 2b, Supplementary Table 3 and Supplementary Fig. 4b). The distance of the Met axial ligand has been associated with an increase in redox potential of the T1Cu centre[38,39]. Other than the distortion of the T1Cu Met ligand, comparison with the lower pH T1Cu(II) coordination reveals minimal differences (Supplementary Fig. 4h). Small differences are reflected in the optical spectrum where the high pH spectrum shows ~2 nm shift of the 450 nm peak, with the ratios between the two peak maxima also decreasing by ~ 10 % (Fig. 3a, b).

## Nitrite-bound oxidised *Br*NiR (pH 5.5) at 1.02 Å resolution

There are some atomic resolution structures of nitrite-bound CuNiR that have been obtained using synchrotron radiation (SR) X-rays. It is now apparent that these synchrotron structures are all affected to some extent by the X-ray radiolysis that occur during the data collection, resulting in a reduction of the T1Cu site and transfer of electrons to the nitrite-bound T2Cu site, initiating the enzymatic turnover. These structures, thus, show a mixed population of substrate, sometimes in more than one conformation and the product NO. Single-shot experiments on CuNiRs have been performed using the femtosecond (-10 fs) X-ray pulses from XFEL to outrun the radiation-induced chemistry. Both types of serial femtosecond crystallography have been used: SFX, which uses a large number of small (<5 μm) crystals and serial femtosecond rotational crystallography (SF-ROX), which utilises fewer larger crystals where the crystal is translated between rotations. The best resolution of nitrite-bound structures of CuNiR using SFX and SF-ROX based has been -1.6 Å[17] and 1.3 Å[16] respectively. We applied 13 keV X-rays from SACLA to obtain an atomic resolution structure of oxidised nitrite-soaked *Br*NiR (pH 5.5), free from radiation-induced chemistry, at 1.02 Å resolution (Supplementary Table 1,2). An optical spectrum of a nitrite-soaked crystal prior to collection of XFEL data, clearly confirmed the presence of nitrite and the Cu(II) oxidation state of the T1Cu centre (Fig. 3b). The structure showed a fully occupied nitrite molecule coordinated to T2Cu(II) (Supplementary Fig. 1c) with equal N-O bond lengths (-1.27 Å; O$^1$-N [1.27 (3)]/O$^2$-N [1.27 (4)], an angle of 131.5° (3)) and a binding mode resembling a "top-hat" conformation (Fig. 2c, f) - where the two oxygen atoms (O$^1$/O$^2$) would be expected to be coordinated in a bidentate binding mode, with no interaction of the nitrogen atom. In this structure, the coordinated nitrite molecule shows asymmetric bidentate binding (O$^2$: 1.96 (2) Å, N: 2.27 (3) Å, O$^1$: 2.25 (1) Å) (Supplementary Table 3 and Supplementary Fig. 2c) with the

oxygen atom (O$^2$) closest to Asp$_{CAT}$ coordinating more favourably to T2Cu(II) than the other two atoms. This observation contrasts with the previous lower resolution structure where a "side-on" nitrite molecule was seen coordinated to the T2Cu(II) with almost equidistant coordination (O$^2$: 1.92 Å, N: 2.02 Å, O$^1$: 2.12 Å)[16]. This structure also reveals the presence of both the proximal conformation of Asp$_{CAT}$ (83 %) and gatekeeper conformation (17 %) (Fig. 2c, f and Supplementary Fig. 1c), which was not seen in the previous 1.3 Å resolution SF-ROX XFEL structure[16]. The higher resolution of 1.02 Å has provided a significant improvement in the quality of the map to resolve the minor conformation. The gatekeeper position has been seen previously in the SR crystallographic structures of *Br*NiR and other prototypic CuNiRs (i.e., *Ac*NiR, *Af*NiR) to varying degrees, but in these cases, multiple conformations of nitrite or nitrite and product are observed. The presence of gatekeeper conformation with full occupancy single conformation of nitrite likely suggests that the "top-hat" conformation of nitrite represents the initial conformation of the binding to T2Cu after delivery of nitrite into the pocket. The proximal/gatekeeper positions of Asp$_{CAT}$ are both present with new partial occupancy waters near the active site cavity: a gatekeeper water (W$_G$; 14 %) which takes the place of the second carboxyl oxygen (O$^{\delta 1}$) atom of Asp$_{CAT}$ in the minor conformation and a proximal water (W$_P$; 20 %), which occupies a similar position to W3 seen in the other structures, with the major conformation. New mobile waters W4a/W4b are also observed closer to the nitrite molecule. The O$^{\delta 2}$ atom of the Asp$_{CAT}$ proximal position bonds favourably with the O$^1$ atom of the T2Cu-bound nitrite molecule (2.34 Å) – this distance is almost identical to the W1-O$^{\delta 2}$ (Asp$_{CAT}$) distance (23.6 Å) in the oxidised structure at pH 5.5. The conserved bridging water (W$_{Br}$), bridges again with Asp$_{CAT}$ and His$_{CAT}$, with His$_{CAT}$ again rotated towards Glu274 with a hydrogen bonding distance of 2.63 Å (Supplementary Fig. 3c). At the T1Cu(II) site, disorder of the Met145 T1Cu ligand is observed once more (Fig. 2c) but with equal proportions this time (ordered (52%; 2.47 (7) Å); disordered (48%; 2.71 (8) Å)) (Supplementary Table 2 and Supplementary Fig. 4c).

## Chemically reduced *Br*NiR (pH 5.5) at 1.05 Å resolution

A reduced form of any *Br*NiR has yet to be structurally characterised and only a few structures in general of CuNiR have been reported for the reduced enzymes[15,40–43]. A cryogenic XFEL structure of the reduced form of *Ac*NiR was previously obtained using the SF-ROX approach at 1.6 Å resolution[15]. To extend on that result and provide fresh insight into the reduced form of a *Br*NiR, an atomic resolution XFEL structure of chemically reduced (by dithionite) *Br*NiR at pH 5.5 was also successfully determined at 1.05 Å resolution, with each crystal soaked with the strong reductant dithionite before cryocooling for SF-ROX data collection. The optical spectrum of a crystal was obtained beforehand to ensure that full reduction of T1Cu(I) had occurred, with full bleaching of T1Cu absorbance peaks observed (Fig. 3a).

At the reduced T1Cu(I) site, marked differences are observed when compared with the equivalent oxidised form (atomic resolution XFEL structure of as-isolated *Br*NiR (pH 5.5) at 1.15 Å resolution). In the reduced form of the enzyme, two positions of the T1Cu(I) are observed (Fig. 4d–f and Supplementary Fig. 4d), representing two reduced states - with the major conformation (77 %) shifting 0.17 Å from its oxidised position (Supplementary Fig. 4g). This is similar to the equivalent *Ac*NiR structure[15], where T1Cu shifted 0.2 Å upon reduction and also revealed two positions of the reduced state of T1Cu. To our surprise, though, in this higher resolution chemically-reduced *Br*NiR structure, the major reduced T1Cu is associated with a classical tetra-coordinated site (Fig. 4e and Supplementary Fig. 4d) but the minor reduced position (23 %) is associated with a unique tricoordinated site (Fig. 4f and Supplementary Fig. 4d), where the S$^\delta$ atom of the coordinating Met145 ligand is completely flipped from the coordination sphere. This was previously only observed in an as-isolated *Br*NiR enzyme at high pH, following prolonged X-ray exposure of the same

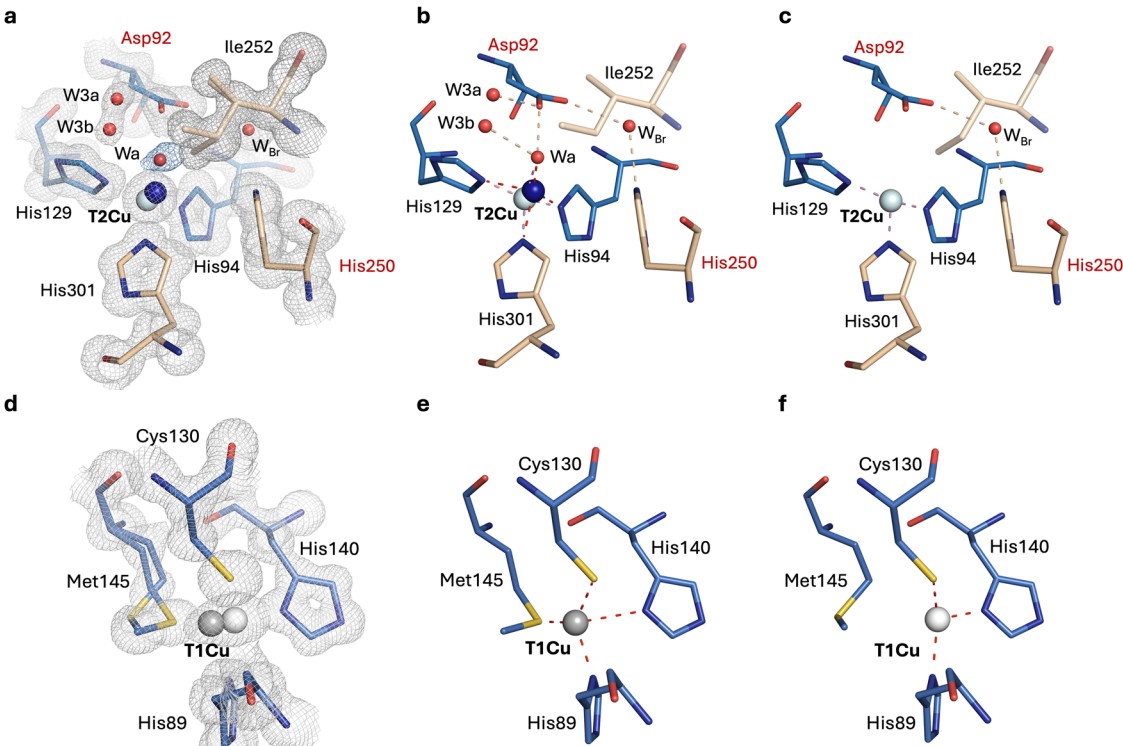

**Fig. 4 | Atomic resolution (1.05 Å) XFEL structure of Chemically-reduced *Br*NiR at pH 5.5. a** T2Cu(I) active site, showing the conformations present and two positions of T2Cu, representing 'reduced' (blue) and fully reduced (cyan). **b** Tetracoordinated active site conformation 1 with partial coordination of water ligand to T2Cu(I) still present. **c** Tricoordinated active site conformation 2 with His₃-T2Cu(I) coordination and flipping of the Ile252 residue into the free space of the cavity. This conformation exists only with the fully-reduced position of T2Cu (cyan). **d** T1Cu(I) redox site, showing two conformations present and two positions of reduced T1Cu(I) (white and grey). **e** Tetracoordinated redox site conformation 1 with no flipping of the coordinating Met145 residue and position 1 of reduced T1Cu (grey). **f** Tri-coordinated redox site conformation 2 with flipping of the coordinating Met145 residue from position 2 of reduced T1Cu (white). Residues from chain A are coloured in blue and chain B residues are coloured in gold. Residues are labelled in black, and catalytic Asp and His residues are labelled in red. Waters are coloured in red. $W_{Br}$ = Bridging water. $2F_o$-$F_c$ density map is contoured to 1 σ and coloured in grey.

crystal volume in an MSOX experiment using a focused X-ray beam at a synchrotron beamline[26].

At the reduced T2Cu(I) site, two active site conformations and positions of T2Cu(I) are observed, representing a tetracoordinated T2Cu(I) site where a solvent ligand (Wa) is retained and a tricoordinated T2Cu(I) site where the solvent ligand (Wa) is absent (Fig. 4a–c and Supplementary Fig. 1d and 2d). The T2Cu drops 0.5 Å into the histidine plane in the major conformation (66 %), with the minor position (34 %) similar to that seen in the other atomic resolution *Br*NiR XFEL structures also present. In the active site cavity, a half-occupancy water ligand (Wa; 54 %) is observed coordinated to T2Cu in one active site conformation (Fig. 4b and Supplementary Fig. 2d), with coordination distances of 1.72 Å and 2.16 (2) Å for the two T2Cu(I) positions, respectively. In the second conformation, the Wa ligand is lost and the site becomes devoid of an external ligand, with the $C^{\delta1}$ side-chain of Ile252 flipping into the available free space (46 %) (Fig. 4c and Supplementary Fig. 1d). The latter is consistent with a single tricoordinated T2Cu(I) site that was observed in the lower resolution chemically reduced SF-ROX structure of *Ac*NiR at 1.6 Å resolution[15]. $Asp_{CAT}$ remains in a single proximal position, with some mobility of the $O^{\delta2}$ atom observed and dual-occupancy water (W3) also present. $His_{CAT}$ is instead rotated towards the hydroxyl oxygen atom ($O^{T1}$) of Thr275, with roughly equal hydrogen bond distances of 2.67 Å and 2.75 Å between the Glu274 and Thr275 residues, respectively (Supplementary Fig. 3 d).

### Accurate atomic resolution XFEL structures of prototypic *Ac*NiR

Cryogenic (100 K) XFEL structures of the prototypic *Ac*NiR in its fully-oxidised as-isolated and nitrite-bound catalytic states were also obtained at pH 4.8, both at resolutions of 0.95 Å (Supplementary Table 1,2). To the best of our knowledge, these sub-atomic (<1 Å) resolution structures represent the highest resolution XFEL structures to date for a macromolecular molecule. The previous cryogenic XFEL structures obtained for *Ac*NiR using the SF-ROX approach were limited to a resolution of 1.5 Å[15]. With the extension of the resolution to ≤1 Å, unrestrained SHELXL refinement could now also be performed for both structures, providing highly accurate bond distance and angle information relevant to the redox chemical reaction in these model biological systems (esds values are provided in brackets).

### As-isolated oxidised *Ac*NiR (pH 4.8) at 0.95 Å resolution

We present the structure of as-isolated oxidised *Ac*NiR (pH 4.8) at a sub-atomic level of resolution of 0.95 Å (Fig. 5). The structure revealed previously unseen details of the T2Cu(II) site architecture that were not observable in the previous much lower resolution (1.5 Å) SF-ROX structure[15]. The higher resolution map has enabled the identification of two distinct active site conformations, as seen in the Fo-Fc OMIT maps (Supplementary Fig. 1e): in one, the fully occupied water molecule seen previously can again be observed in a distorted geometry relative to the histidine plane (i.e., in a position reminiscent of W2 denoted for *Bradyrhizobium* CuNiRs). However, in this conformation, an additional partial T2Cu(II) water molecule (W1; 35 % occupancy) is now also observed, revealing a fractional penta-coordinated oxidised state (Fig. 5c) for a prototypic CuNiR (i.e., *Ac*NiR, *Ax*NiR and *Af*NiR). The partial water is positioned similarly to W1 of *Bradyrhizobium* CuNiRs and is hydrogen bonded to W2 (2.32 Å) but is coordinated to T2Cu by an unusually short distance (1.84 (2) Å)

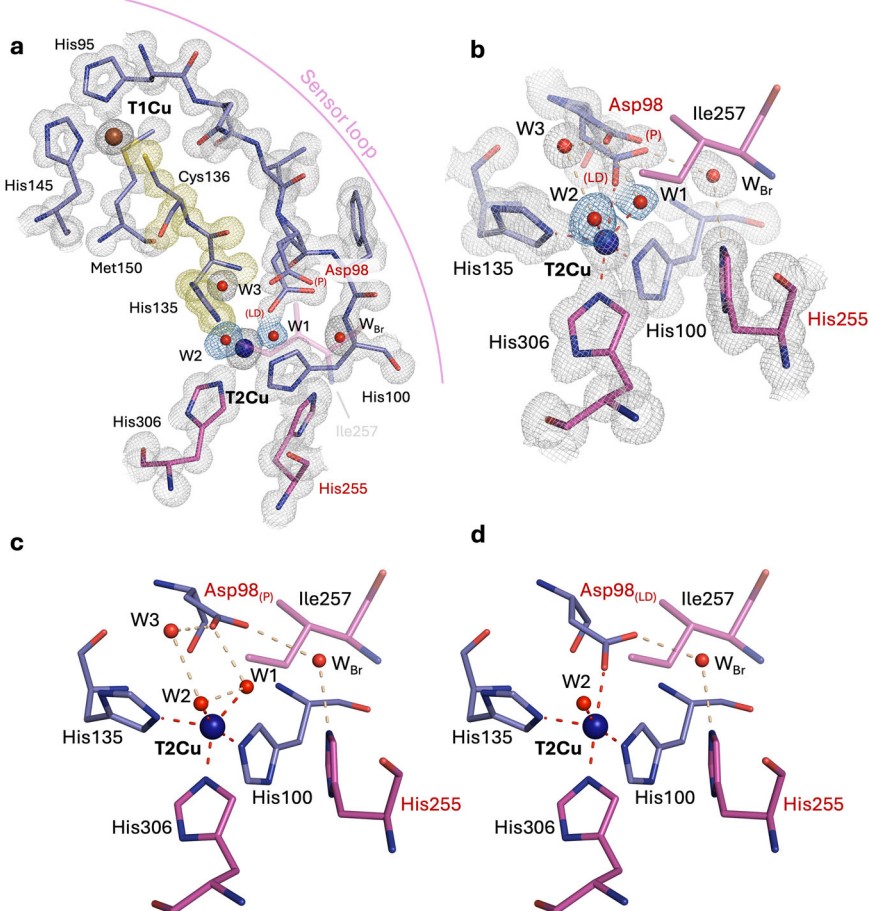

**Fig. 5 | Atomic resolution (0.95 Å) XFEL structure of prototypic CuNiR from *Achromobacter cycloclastes* (*Ac*NiR) in its as-isolated state at pH 4.8. a** T1Cu and T2Cu metal centres linked via a conserved Cys-His bridge (yellow density) and the sensor loop. **b** T2Cu active site, showing two conformations present: **c** Active site conformation 1 with coordination of two water ligands to T2Cu, showing a previously unseen pentacoordinated T2Cu(II) with two solvent ligands for a prototypic CuNiR, coordination involves a full-occupancy solvent ligand (W2) and partial-occupancy water (W1). **d** Active site conformation 2 with coordination of a water ligand to T2Cu and direct coordination of the catalytic Aspartic residue to T2Cu, not observed previously. Residues from chain A are coloured in purple, and chain B residues are coloured magenta. Waters are coloured in red. LD lower distorted Asp$_{CAT}$ conformation, P proximal Asp$_{CAT}$ conformation. W$_{Br}$ Bridging water. 2F$_o$-F$_c$ density map is contoured to 1 σ and coloured in grey (blue for T2Cu ligands and yellow for Cys-His bridge).

compared to W2 (2.03 (1) Å), which may indicate a hydroxide (OH⁻) ligand. The second T2Cu(II) active site conformation also reveals a completely unique architecture, extending from the observations of the previous structure, which revealed the presence of a lower distorted proximal conformation of Asp$_{CAT}$ (Asp98).

In the current atomic resolution XFEL structure, this lower distorted conformation of Asp$_{CAT}$ (65% occupancy), is rotated 38° around the O$^{δ2}$ atom, facing much lower towards the Cu atom and consequently allowing the O$^{δ2}$ atom to be positioned only 2.20 (2) Å away from the T2Cu(II), a distance indicating direct interaction with T2Cu(II) (Fig. 5d). It occupies the site with W2 when W1 is not present – in 2 monomers/homotrimer. When this unique T2Cu coordination of Asp$_{CAT}$ is not observed, the classical proximal position is retained together with W1 (35 %) and W2 at the T2Cu(II) site. W3 is also observed within the active site once more, linking with the fully occupied W2 ligand and proximal Asp$_{CAT}$ conformation. His$_{CAT}$ is rotated towards Thr280 in this structure, ensuring suitable distances for hydrogen bonding to both Thr280 (2.77 Å) and Glu279 (2.67 Å).

At the T1Cu(II) redox centre, the tetracoordinated site is consistent with the previous XFEL structure, with coordination distances (Supplementary Table 2 and Supplementary Fig. 4e) in agreement with a fully-oxidised redox state as also confirmed by the single crystal optical spectrum (Fig. 3c). Slight shortening of the Met-T1Cu(II)

distance and elongation of the Cys-T1Cu(II) is observed, common for the green subset of T1Cu proteins. Comparison with the oxidised T1Cu(II) of *Br*NiR, reveals a slight shift in geometry as a result (Supplementary Fig. 4i).

## Nitrite-bound oxidised *Ac*NiR (pH 4.8) at 0.95 Å resolution

The sub-atomic resolution XFEL structure of nitrite-bound oxidised *Ac*NiR (pH 4.8) was determined to 0.95 Å resolution (Supplementary Table 1,2). The single crystal optical spectroscopy prior to collection of XFEL data confirmed that the enzyme was in the oxidised Cu(II) state (Fig. 3c). When compared to the previously collected lower resolution cryogenic XFEL structure (1.5 Å resolution)[15], several previously unseen features are observed. In the 0.95 Å resolution structure (Fig. 6a, b), two binding modes of the nitrite were observed at the T2Cu(II) active site with compatible Asp$_{CAT}$ conformation (proximal/gatekeeper) (Fig. 6b), with these evident in the Fo-Fc OMIT maps (Supplementary Fig. 1f). The higher resolution structure, combined with SHELXL refinement, shows two variations of the "top-hat" binding mode, as opposed to a "top-hat" and "side-on" binding mode previously modelled in the lower resolution SF-ROX structure. The major occupancy "top-hat" nitrite$_1$ conformation (65 % occupancy) (Fig. 6c) has almost equal N-O bond lengths (-1.28 Å; O$^1$-N [1.29 (3)]/O$^2$-N [1.28 (3)]) and an angle of 111° (2)) and exhibits bidentate coordination to the T2Cu(II) via

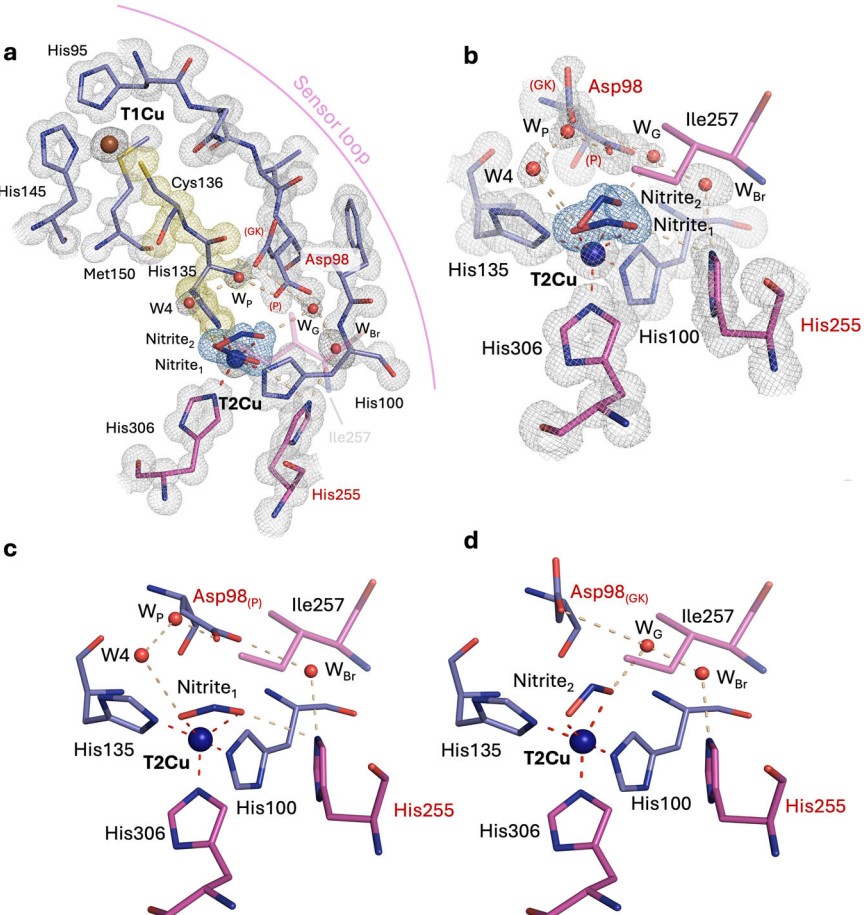

**Fig. 6 | Atomic resolution (10.95 Å) XFEL structure of prototypic CuNiR from *Achromobacter cycloclastes* (*Ac*NiR) in its nitrite-bound state at pH 4.8. a** T1Cu and T2Cu metal centres linked via a conserved Cys-His bridge (yellow density) and the sensor loop. **b** T2Cu active site, showing two conformations of 'top-hat' nitrite present: **c** Active site conformation 1, showing first nitrite 'top-hat' conformation, with preferred bonding to the His$_{CAT}$ residue and proximal position of Asp$_{CAT}$ (**d**) Active site conformation 2, showing second nitrite 'top-hat' conformation, with preferred bonding to a new water that appears when Asp$_{CAT}$ is in a gatekeeper position, referred to as gatekeeper water (W$_G$). Residues from chain A are coloured in purple, and chain B residues are coloured magenta. Waters are coloured in red. GK Gatekeeper Asp$_{CAT}$ conformation, P proximal Asp$_{CAT}$ conformation, W$_{Br}$ Bridging water, W$_G$ gatekeeper water, W$_P$ proximal water 2F$_o$-F$_c$ density map is contoured to 1 σ and coloured in grey (blue for T2Cu ligands and yellow for Cys-His bridge).

the two oxygen atoms (O$^1$: 1.88 (2) Å, N: 2.42 (3) Å, O$^2$: 2.14 (2) Å) (Supplementary Table 3 and Supplementary Fig. 2f). This conformation correlates with the proximal position of Asp$_{CAT}$ (68 % occupancy) due to active site steric restraints but the O$^2$ atom of the molecule is more suitably positioned to the N$^{ε2}$ atom of His$_{CAT}$ (3.06 Å) for hydrogen bond formation compared to Asp$_{CAT}$. This active site conformation occupies ~ 2 monomers/homotrimer. Other partial waters are also observed in this active site conformation (W$_P$ and W4) with these linking to the nitrite molecule.

The minor occupancy "top-hat" nitrite$_2$ conformation (35 % occupancy) (Fig. 6d), ~1 monomer/homotrimer, also has almost equal N-O bond lengths (~1.27 Å; O1-N [1.28 (3)]/O2-N [1.27 (3)]) and an angle of 120° (2)) and again exhibits bidentate coordination to the T2Cu(II) via its two oxygen atoms (O$^1$: 2.11 (2) Å, N: 2.40 (2) Å, O$^2$: 2.04 (1) Å) (Supplementary Table 3 and Supplementary Fig. 2f). This conformation now correlates with the gatekeeper position of Asp$_{CAT}$ (32 % occupancy) and a new partial occupancy water (W$_G$; 33 %) which takes the place of the second carboxyl oxygen (O$^{δ2}$) atom of Asp$_{CAT}$, which is at a more realistic distance for hydrogen bond formation with the O$^2$ of nitrite than Asp$_{CAT}$ (2.71 Å compared to 2.01 Å). The observation of two conformations of nitrite, both with a "top-hat" binding mode, alongside two conformations of Asp$_{CAT}$, provides persuasive evidence that the "top-hat" binding mode is the initial binding mode of nitrite before the beginning of catalysis – in

contrast with our previous suggestion of a "side-on" being the initial coordination mode[16]. His$_{CAT}$ is similarly rotated towards Thr280 like the as-isolated *Ac*NiR structure, with suitable distances for hydrogen bonding to both Thr280 (2.77 Å) and Glu279 (2.70 Å).

At the T1Cu(II) redox centre, distances for the tetracoordinated site are almost identical to the as-isolated oxidised state (Supplementary Table 2 and Supplementary Fig. 4e). Comparison with the equivalent state in the bluish-green *Br*NiR, revealed a similar shift in geometry as the *Br*NiR and *Ac*NiR as-isolated oxidised species and confirmed that the slight disorder of the Met ligand observed in the tetracoordinated T1Cu(II) site in *Br*NiR is representative of a mixture of blue and green Met positions (Supplementary Fig. 4j).

**Single crystals spectroscopy of copper-nitrite reductase species**
Single crystal spectroscopy at 100 K on the same crystalline sample was also performed prior to SF-ROX measurements at SACLA to confirm the valence state of the T1Cu redox centre. Figure 3 shows the optical spectra from single crystals of bluish-green CuNiRs from two *Bradyrhizobium* species (*Bradyrhizobium sp. ORS 375* referred to as *Br$^{2D}$*NiR (Fig. 3a) or *Bradyrhizobium japonicum* referred to as *BrJ*NiR (Fig. 3b)) at pH 5.5 and pH 7.3 and prototypic green CuNiR, *Ac*NiR at pH 4.8 (Fig. 3c). For the as-isolated crystals of each CuNiR, after soaking with same buffer conditions and cryoprotectants as that used for the SF-ROX measurements, confirmed for all cases the T1Cu redox centre were in

their fully oxidised Cu(II) states. For comparison, spectra for solution $Br^{2D}$NiR at pH 6.5 are also included (Fig. 3d). In all cases, optical spectra of the oxidised enzyme show the bands that have been reported for a variety of CuNiRs that give rise to their colours, blue, green, and bluish-green[11]. In comparison to the solution spectra at room temperature (Fig. 3d), single crystal spectra taken at -100 K are much sharper, where some of the weaker broad bands become well resolved. For example, the broad feature between 650–800 nm that has been assigned to d-d transition, becomes well resolved bands at -680 nm and -770 nm. Likewise, the shoulder at -380 nm, assigned to a S(Met) → Cu ligand-to-metal charge transfer (LMCT), resolves into a clear peak. Similar sharpening has been seen for CuNiR from *Rhodobacter sphaeroides*, (*Rs*NiR) for solution samples when cooled to 7 K[10]. Relative intensities of -570 nm, assigned to S(Cys) → Cu π CT transition and -460 nm, assigned to S(Cys) → Cu σ CT band, give rise to the colour with *Ac*NiR being green and *Bradyrhizobium* CuNiRs (*Br*²DNiR and *Brj*NiR) being bluish-green. Addition of nitrite to the crystals of oxidised enzymes (with the same soaking procedure as XFEL data collection) shows a band associated with a LMCT band of nitrite at -350 nm. It is clearly resolved in the single crystal spectra of both *Bradyrhizobium* species, but for *Ac*NiR it overlaps with the 380 nm band. Upon chemical reduction of the oxidised enzyme crystals, all the optical bands for these CuNiRs are bleached, consistent with T1Cu being in Cu(I) oxidation state. The addition of nitrite to these reduced crystals results in a small recovery of the optical bands, irrespective of the duration of the soak. We note that the length of the long soak was significantly more than the soaking time used for soaking nitrite to the oxidised crystals that yielded full occupancy of nitrite at the catalytic T2Cu site. The low recovery of T1Cu optical spectrum (-20 %) indicates that binding of nitrite to the reduced T2Cu(I) site is limited but not prevented, consistent with earlier observations on CuNiR from *Alcaligenese xylosoxidans* (*Ax*NiR)[33], providing clear evidence for the binding-before-reduction pathway as opposed to the reduction-before-binding pathway[34], which also agrees with the structural observations of the mixed T2Cu(I) sites that are observed here in the atomic resolution XFEL structures of the dithionite-reduced *Br*NiR, with one maintaining a water ligand capable of binding nitrite via displacement of the water and the consequent reoxidation of the T1Cu resulting from turnover, and the other representing the inactive dead-end (T2Cu(His₃) state.

### Protonation states for the catalytic aspartic and histidine residues from SHELXL-refined accurate atomic resolution single-shot XFEL structures

With the cardinal combination of atomic resolution damage-free structures obtained on crystals whose oxidation state has been validated by cryogenic single crystal spectroscopy prior to single-shot XFEL pulses (<10 fs) Sf-ROX data collection, together with SHELXL refinement, a unique opportunity is provided to assess the protonation states of the two catalytic residues in these CuNiRs ($Asp_{CAT}$ and $His_{CAT}$) during some of the key catalytic states. This is enabled by a final step of unrestrained refinement to remove the influence of crystallographic dictionary restraints and for estimation of standard uncertainties (esds) of bond lengths and angles. We originally applied a similar method for atomic resolution synchrotron structures of the *Br*NiR from *Bradyrhizobium sp. ORS 375*, using predication methods previously suggested, where in short: data from the Cambridge Structural Database (CSD)[44] has suggested that for aspartate residues the bond lengths between Cγ–O⁶¹ and Cγ–O⁶² are expected to be equal to - 1.256 Å when non-protonated and when protonated one bond will increase to - 1.310 Å and the other will become a double bond with a distance of - 1.210 Å[45]. For histidine residues, the bond angles are used to assign its protonation state and the differences between protonated and non-protonated states are relatively small, so greater caution is required. An analysis of imidazole moieties in histidine residues from atomic resolution structures

in the CSD and the Protein Data Bank (PDB)[46] has suggested that histidine is fully protonated if the bond angles of Cᵉ¹-Nᵉ²-C⁶² and Cγ-N⁶¹-Cᵉ¹ are greater than 108.9°[47]. In our current analysis, we estimate that $His_{CAT}$ is protonated if the bond angles of Cᵉ¹-Nᵉ²-C⁶² and Cγ -N⁶¹-Cᵉ¹ are greater than 107°. In contrast to the electron density maps obtained in the atomic resolution structures using storage ring X-rays[48], no hydrogen atoms were observed in neither $2F_o$-$F_c$ or $F_o$-$F_c$ maps, presumably because of the pulsed nature of XFEL (pulse to pulse variations) and large number of crystals, each with multiple spots, used for data collection. Despite the lack of guidance from the experimental density maps, the analysis of the Cᵉ¹-Nᵉ²-C⁶² and Cγ -N⁶¹-Cᵉ¹ angles of $His_{CAT}$ (Supplementary Table 4) allowed us to conclude that this residue is likely to be fully protonated in all structures. There are differences in $Asp_{CAT}$ protonation depending on the catalytic state of the protein. In the pH 5.5 oxidised *Br*NiR structure, the O⁶²-Cγ 1.27 (2) Å bond length is slightly longer than O⁶¹-Cγ: 1.25 (2) Å, which may be due to light protonation on O⁶², making it almost neutral. In the dithionite-reduced *Br*NiR structure (pH 5.5) the O⁶²-Cγ: 1.29 (2) Å is more elongated, which indicates the potential full protonation on O⁶² and thus is not negatively charged. For the high pH oxidised (pH 7.3) and nitrite-bound (pH 5.5) *Br*NiR structures, $Asp_{CAT}$ appears to be non-protonated (both bond lengths are equal), or negatively charged. In *Ac*NiR, for both the nitrite-bound (pH 4.8) and the oxidised (pH 4.8) forms, $Asp_{CAT}$ in its proximal conformation appears to be non-protonated and negatively charged. However, in the lower distorted $Asp_{CAT}$ conformation in the oxidised structure (pH 4.8), there is a large asymmetry in the bond lengths of $Asp_{CAT}$ (O⁶¹-Cγ: 1.24 (3) Å, O⁶²-Cγ: 1.34 (4) Å), which would indicate O⁶² protonation, with O⁶² creating the direct bond with T2Cu at a distance of 2.2 (1) Å. Supplementary Tables 4 and 5 summarises the bond lengths, angles and protonation states for these catalytic residues.

## Discussion

We have applied the serial femtosecond rotational crystallography (SF-ROX) approach[49], where crystallographic data have been collected in an automated manner, allowing the method to be time-efficient during time-limited XFEL beamtime. We demonstrate that the ability of XFEL to deliver higher energy X-rays >13 keV of around ten femtosecond pulses can be harnessed to provide atomic resolution structures that are free from X-ray induced redox or chemical changes in the biological system under investigation. Such high resolution is currently achievable on crystals with - 30 μm thicknesses, making SF-ROX an ideal approach.

The atomic resolution XFEL structures of CuNiRs, together with refinement of ADP's (Supplementary Fig. 5) and a final cycle of unrestrained refinement in SHELXL, have provided the most accurate structures to date for a macromolecular and metal-containing system. More importantly, catalytically relevant states are captured before radiation-induced chemistry can occur. The optimal static structures collected at 100 K reveal features that were previously undetected in equivalent lower resolution XFEL structures also obtained at 100 K[15,16] and equivalent cryogenic atomic resolution structures obtained at synchrotron radiation (SR) sources[14,16,26,27,50,51]. The structures provide a consistent picture of the Cu centres and residues in the catalytic pocket for both *Bradyrhizobium* CuNiRs and the prototypic *Ac*NiR, which were previously shown to have differences in their lower resolution XFEL structures and higher resolution SR structures. These differences were previously assigned as explanations for marked differences in their catalytic activity[16,26,28]. Here, the atomic resolution XFEL structures of both *Ac*NiR and *Br*NiR in the oxidised state show the catalytic T2Cu(II) to be penta-coordinated, with a (His)₃ coordination and two solvent ligands, either two waters or a water and a hydroxide. This consistent picture is summarised in the Fig. 7, where key findings from the current studies are provided.

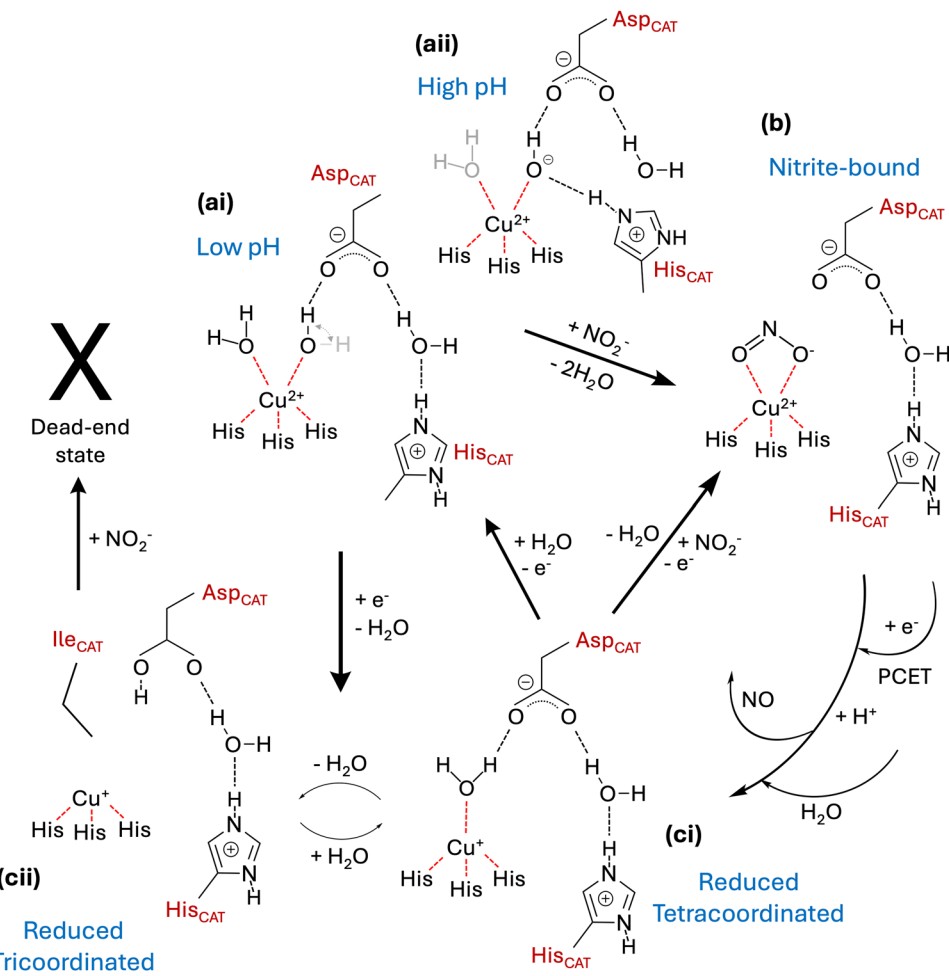

**Fig. 7 | A consistent enzymatic scheme is presented from the combination of true atomic resolution XFEL structures, single crystal spectroscopy and SHELXL analysis of *Bradyrhizobium* CuNiRs (*Br*NiR) and *Achromobacter cyclo-clastes* CuNiR (*Ac*NiR).** The resting state of the oxidised enzyme (as confirmed by single crystal spectroscopy) at both low (ai) and high (aii) pH exhibits a penta-coordinated catalytic type 2 Cu centre. At high pH, the bridge between His$_{CAT}$ and Asp$_{CAT}$ is lost, with His$_{CAT}$ bridging to one of the T2Cu-coordinated solvent ligands - water/hydroxide. Reduction leads to two conformations, ci and cii, where the cii conformation, previously unobserved, is likely to be capable of binding nitrite through the displacement of a single water. Such a state is required as part of the random-sequential scheme but has never been structurally observed. Nitrite-bound species (b) of the oxidised enzyme is formed by displacement of two solvent ligands, which, following a proton-coupled electron transfer (PCET) reaction[26,28], would form the product nitric oxide (NO) species[16].

In the nitrite-bound structures, both solvent-derived ligands are displaced, but the pentacoordinated T2Cu(II) is retained, with bidentate coordination of the two oxygen atoms from nitrite in a "top-hat" binding mode. Pentacoordination has been observed previously in the oxidised resting state, particularly for the lower activity *Bradyrhizobium* CuNiRs, with the additional coordinated water originally suggested to be responsible for its lower catalytic efficiency owing to thermodynamically unfavourable displacement of the waters with nitrite[16,23]. As pentacoordination of an oxidised T2Cu(II) site now appears to be a common feature for all CuNiRs, a more likely explanation for the lower catalytic activity of *Br*NiRs is likely due to a slower rate of inter-Cu ET, which has been strongly supported by our kinetic studies[28]. At higher pH (i.e., pH 7.3 collected here), the partial occupancy of the weakly coordinating water could suggest a tetra-coordinated T2Cu(II) site may still exist under these conditions, where activity is known to be severely disrupted[35,37].

The predicted protonation states of His$_{CAT}$ and Asp$_{CAT}$ are consistent with the former being fully protonated in all the structures we determined, in contrast to Asp$_{CAT}$, which shows different protonation states. Multiple structures from one crystal (MSOX) movies show that the Asp$_{CAT}$ residue is mobile during catalysis and the atomic resolution XFEL structures presented here identify a previously unseen lower distorted conformation of this residue in the resting state *Ac*NiR that displaces W1 to become a ligand to the T2Cu. In this conformation, it remains H-bonded to the bridging water between Asp$_{CAT}$ and His$_{CAT}$. This unusual direct coordination of Asp$_{CAT}$ to T2Cu maintains a five-coordinated T2Cu(II) when the second solvent ligand is not present. This previously unseen conformation is lost on nitrite binding and depending on the nitrite conformation, Asp$_{CAT}$ then assumes proximal and gatekeeper positions with the latter H-bonding to W$_G$, a second bridging water between Asp$_{CAT}$ and His$_{CAT}$. In the "top hat" binding mode, nitrite can form a direct H-bond with His$_{CAT}$ if suitably positioned, being poised for catalysis following proton uptake and inter-Cu-ET, which we have shown occurs at the same rate, with the proto-nation of Asp$_{CAT}$ gating ET[52]. His$_{CAT}$ has also been shown to act as a redox-coupled proton switch, with inter-Cu ET from T1Cu to T2Cu facilitating a rotation of its imidazole ring shifting hydrogen bonding of its N$^{\delta 1}$ atom from the stronger carbonyl O atom of Glu274(279) bond to the less negatively charged and weaker bond of Thr275(279), thus promoting proton transfer to the bridging water or directly to the closest T2Cu ligand atom via its N$^{\varepsilon 2}$ atom[17]. Different rotamers of His$_{CAT}$ were observed for the oxidised structures of *Br*NiR and *Ac*NiR, with rotation observed only for *Ac*NiR. The rotamers are also in agreement with the previous lower resolution 100 K SF-ROX structures

of BrNiR[16] and AcNiR[15] with crystals crystallised at the same pH values (AcNiR was pH 5 previously).

The higher accuracy of metrical information achieved here also allows us to assess if the two water molecules coordinated to T2Cu(II) differ in any way. For both enzymes at low pH, where the activity is optimal, one of the coordinated waters is quite short suggesting a potential water – hydroxide ($OH^-$) resting state of the enzymes in these structures. Furthermore, in the high pH structure, where it would be expected to accommodate a deprotonated water ligand or hydroxide ligand, shows the direct bridging network of $Asp_{CAT}$ - $OH^-$ ligand - $His_{CAT}$, completely omitting the conserved bridging water. The presence of a $OH^-$ T2Cu ligand in the resting state of the enzymes has been proposed previously in high-resolution neutron crystallography and computational studies[53,54]. The position of the $OH^-$ ligand is in an optimal position to receive a proton from either the $Asp_{CAT}$, the bridging water or $His_{CAT}$ and is preferably coordinated to T2Cu. The fully-occupied water coordinated together with the hydroxide is likely more energetically poised for displacement upon nitrite binding or following reduction of the T2Cu, to leave a tetracoordinated site with tetrahedral geometry. For the prototypic blue AxNiR, measurements of proton consumption during catalytic turnover at pH 7.0 found that 2 protons derived from the solvent were consumed during a single turnover, indicating $H_2O$ rather than $OH^-$ was the ligand to the T2Cu in the resting state[24]. This assignment was confirmed by our neutron crystallographic structural data for AcNiR[15].

The atomic resolution XFEL structure of dithionite-reduced BrNiR showed significant changes in both Cu centres. On reduction, both T1Cu(I) and T2Cu(I) shift by -0.2 Å and 0.5 Å, respectively, with two distinct reduced positions for each copper type. The atomic resolution of the current structure also reveals a minor unique tricoordinated reduced T1Cu(I) (23 %), where the $S^\delta$ atom of the coordinating Met145 ligand is completely flipped from the coordination sphere. The reduced T2Cu(I) site eventually becomes devoid of solvent-derived ligands, with the $C^{\delta1}$ sidechain of Ile252 flipping to occupy the available free space. The latter is consistent with a fully-reduced tricoordinated T2Cu(I) site observed in AcNiR and represents the non-reactive dead-end state incapable of binding nitrite for further turnover[33]. For BrNiR (where EPR measurements show that the T2Cu is fully reduced by dithionite), the structure also showed a 50/50 occupancy of a single coordinated solvent ligand (54 %) and only partial formation of the dead-end state (46 %). The observation of tetracoordinated T2Cu(I), despite the use of the strong reductant, suggests this may be the 'reduced T2Cu(I)' site likely to be capable of binding nitrite in the 'reduction before binding' mode of the random sequential mechanism[34]. As tricoordinated T2Cu(I) can only be associated with a flattened T2Cu(I), it means the tetracoordinated T2Cu(I) may exist in either a tetrahedral coordination or square planar coordination, with the square planar coordination transitioning to an empty trigonal planar site. The slower rate of inter-Cu ET in the Bradyrhizobium CuNiRs has allowed these modes to be captured structurally upon chemical reduction, with this previously hidden in the more active AcNiR. Several resting state structures of CuNiRs collected using SR over the years, revealing a four-coordinated T2Cu, may in fact also be a T2Cu(I) species. The optical spectrum of the dithionite-reduced single crystals for all three enzymes confirmed that the T1Cu was in the Cu(I) oxidation state in all instances. Single crystal spectroscopy has also provided clear information on the oxidation states of T1Cu for the same crystalline sample that was used to collect XFEL crystallographic data. The optical spectra confirmed full T1Cu(II) oxidation of the resting and nitrite-bound states. Addition of nitrite to the reduced Cu(I) form of these enzymes in their crystalline state also provided insight into the consequence of the binding of nitrite to the reduced T2Cu(I) site. The low recovery of the T1Cu optical spectrum (-20 %) demonstrated that the binding of nitrite to the reduced T2Cu(I) site is limited but confirms that regular turnover is still accomplished

following the inter-Cu electron transfer to the catalytic site. These results are consistent with the binding-before-reduction branch of the proposed "random sequential mechanism" during turnover[33,34].

We conclude that the ability to obtain true atomic resolution structures, free from radiation-induced chemistry, for a variety of functionally relevant states using few (50) crystals of -30 μm thickness by deploying 13 keV X-rays from an XFEL in conjunction with the SF-ROX methods holds a promise for elucidating the enzyme mechanism for many metalloenzymes. The SF-ROX method, like SFX, is also applicable for ambient temperature crystallography, but the propagation length for damage resulting from the first femtoseconds pulse is going to be significantly larger, thus requiring a much larger number of crystals compared to those used here. The extent of propagation that may occur at room temperatures requires a systematic study.

## Methods

### Expression and purification of recombinant BrNiR and AcNiR
Expression and purification of recombinant BrNiR species into Escherichia coli (E. coli) were performed according to established procedures for BrJNiR[26,32] and $Br^{2D}$NiR[16,28]. Likewise, expression and purification of recombinant AcNiR into E. coli was performed using established procedures[15]. Samples were buffer exchanged into a crystallisation buffer consisting of 10 mM HEPES, pH 6.5 (BrNiR) or 10 mM MES, pH 6.5 (AcNiR) prior to crystallisation.

### Crystallisation of BrNiR and AcNiR and preparation for XFEL
Crystals of BrNiR types and AcNiR were grown using the vapor diffusion hanging drop method at room temperature with 400 μL reservoir solution consisting of 1.8 M ammonium sulphate, 50 mM HEPES (pH 5.0) for BrNiR ($Br^{2D}$NiR); 1.8 M ammonium sulphate, 100 mM Tris (pH 7.3) for BrNiR (BrJNiR); and 1.2 M ammonium sulphate, 50 mM citrate (pH 4.8) for AcNiR, respectively. For BrNiR, ~ 25 mg/mL protein solution was mixed 2:1 with reservoir solution. Crystals grew after a few days - for BrNiR ($Br^{2D}$NiR), cubic-shaped crystals grew in space group $P2_13$ and for BrNiR (BrJNiR), crystals grew in space group $P6_3$. For AcNiR, ~ 20 mg/mL protein solution was mixed 1:2 ratio with the reservoir solution and pyramidal-shaped crystals grew after a few days in $P2_13$ space group. BrNiR crystals were transferred to a storage solution of 2.5 M ammonium sulphate, 50 mM HEPES (pH 5.5) and AcNiR crystals were transferred to a storage solution of 2.5 M ammonium sulphate, 50 mM citrate (pH 4.8) before preparation for XFEL experiments.

For the as-isolated (resting state) structure of BrNiR ($Br^{2D}$NiR) at pH 5.5–42 individual crystals ranging from 400 $μm^3$–700 $μm^3$ were prepared for XFEL collection with crystals soaked in a cryoprotectant solution consisting of 3.3 M ammonium sulphate, 50 mM HEPES (pH 5.5) and 20.3 % sucrose prior to cryocooling by plunging into liquid nitrogen. For the high pH as-isolated (resting state) structure of BrNiR (BrJNiR) at pH 7.3 and nitrite-bound structure of BrNiR (BrJNiR) at pH 5.5 – 64 individual crystals ranging from 500 $μm^3$–700 $μm^3$ were prepared for XFEL collection, respectively. For the high pH BrNiR (BrJNiR) resting state, crystals were soaked in a cryoprotectant solution consisting of 2.8 M ammonium sulphate, 50 mM Tris (pH 7.3) and 17.5 % sucrose for roughly two minutes to ensure successful pH increase prior to cryocooling by plunging into liquid nitrogen. For the nitrite-bound BrNiR (BrJNiR) state, crystals were soaked in a nitrite soaking solution consisting of 2.8 M ammonium sulphate, 50 mM HEPES (pH 5.5) and 200 mM sodium nitrite before being transferred to a cryoprotectant solution consisting of 2.8 M ammonium sulphate, 50 mM HEPES (pH 5.5), 200 mM sodium nitrite and 17.5 % sucrose prior to cryocooling by plunging into liquid nitrogen. For the chemically-reduced structure of BrNiR ($Br^{2D}$NiR) at pH 5.5–64 individual crystals ranging from 400 $μm^3$–500 $μm^3$ were prepared for XFEL collection with crystals soaked in a cryoprotectant solution consisting of 2.8 M ammonium sulphate, 50 mM HEPES (pH 5.5),

100 mM dithionite and 17.5 % sucrose prior to cryocooling by plunging into liquid nitrogen. Crystals changed colour from bluish-green to colourless during soaking, indicating successful reduction of the T1Cu from $Cu^{2+}$ to $Cu^{+}$. 1 M stock solution of HEPES was titrated with HCl to pH 5.5 before mixing with other chemicals for preparation of crystallisation solutions and soaking solutions.

For the as-isolated (resting state) structure of AcNiR at pH 4.8 and the nitrite-bound structure of AcNiR at pH 5.0 − 48 individual crystals ranging from 400 $\mu m^3$–500 $\mu m^3$ were prepared for XFEL collection, respectively. For the resting state, crystals were soaked in a cryoprotectant solution consisting of 2.8 M ammonium sulphate, 50 mM citrate (pH 4.8) and 17.5 % sucrose prior to cryocooling by plunging into liquid nitrogen. For the nitrite-bound state, crystals were soaked in a nitrite soaking solution consisting of 2.5 M ammonium sulphate, 50 mM citrate (pH 4.8) and 200 mM sodium nitrite before being transferred to a cryoprotectant solution consisting of 2.5 M ammonium sulphate, 50 mM citrate (pH 4.8), 200 mM sodium nitrite and 17.5 % sucrose prior to cryocooling by plunging into liquid nitrogen.

### Automatic data collection pipeline of XFEL data at SACLA
All data were collected using higher photon energy (13 keV) at SACLA to push the achievable resolution. Collection was performed on BL2/EH3 at SACLA at 100 K using the serial femtosecond rotational crystallography (SF-ROX) technique[49]. The diffraction images were collected on an MX300-HS detector (Rayonix) with a camera length of 90 mm and a pixel size of 78.2 μm. As-isolated (resting state) data of BrNiR ($Br^{2D}$NiR) at pH 5.5 were collected before the implementation of the automatic data collection pipeline of XFEL data at SACLA. Here, data collection of 41/42 crystals prepared was performed using the SF-ROX technique[49]. The following parameters of XFEL pulses were used: pulse duration (<10 fs), pulse energy (~232 μJ), flux (~4 ×$10^{12}$ photons), beam size at sample position (3.91 μm (H) by 4.14 μm (V)). The crystals were rotated 0.15° and translated 50 μm between each shot and a silicon attenuator with a thickness of 0.22 mm was used.

For all other data sets, the automatic data collection pipeline was used. Here, briefly: four unipucks are loaded into the sample exchange robot (SPACE-II)[55] [each individual unipucks must have the same loop sizes], the unipuck information is updated on the beamline scheduling software (BSS)[56], for each loop size a set number of points are determined, together with the rotation range (e.g. for a 0.5 mm loop: 12 (vertical) x 15 (horizontal) with a rotation range of 1.5° was used (Number of horizontal points x 0.1° oscillation)). The camera distance is selected, and an auto-schedule is generated, which then automatically performs sample centring and collection for each unipuck using the SF-ROX approach. The automatic collection of one unipuck (i.e., 16 crystals) took ~ 1.5 hours. The following parameters of XFEL pulses were used for all: photon energy (13 keV), pulse duration (<10 fs), pulse energy (~205 μJ), flux (~4 × $10^{12}$ photons), beam size at sample position (3.49 μm (H) by 4.14 μm (V)). The crystals were rotated 0.1° and translated 50 μm between each shot. All diffraction images were collected on a MX300-HS detector (Rayonix) with a camera length of 89.2 mm and a pixel size of 78.2 μm and an aluminium attenuator with a thickness of 0.2 mm was used (corresponding to 54 % transmission at 13 keV) throughout.

### Data processing and unrestrained refinement
On-the-fly hit finding was performed using the SACLA processing pipeline, which incorporates Cheetah[57]. Data processing was performed using CrystFEL[58] version 0.9.1 with spot finding and indexing performed using the peakfinder8 algorithm and XGandalf[59], DirAx[60] and Mosflm[61], respectively in indexamajig, with the following parameters (--threshold=10, --min-pix-count=2, --min-snr=15, --local-bg-radius=3). The inner, middle, and outer integration radii were set to 6, 7, and 8 pixels, respectively. Indexing ambiguity was resolved using AmbiGator using the operators k, h, -l for both $P2_1 3$ (m-3)

crystals and $P6_3$ (6/m) crystals. Bragg intensities were merged using the Monte Carlo method with process_hkl, with the high-resolution limit ($d^{-1}$) extended by 2.3 $nm^{-1}$, 3.0 $nm^{-1}$, 3.5 $nm^{-1}$, 4.0 $nm^{-1}$ using the --push-res option for low-pH BrNiR, high-pH BrNiR and low-pH AcNiR, nitrite-bound BrNiR and chemically-reduced BrNiR, and nitrite-bound AcNiR, respectively.

For refinement of anisotropic displacement parameters (ADPs), SHELXL[30] was used. Reflection files in MTZ format and template starting models in PDB format (PDB codes: 6ZAS (BrNiR ($Br^{2D}$NiR)), 8R8S (BrNiR (BrJNiR)) and 6GSQ (AcNiR)) were first converted to HKLF4 and INS SHELX formats using mtz2hkl[62] and pdb2ins[63] programmes, respectively. Several rounds of restrained refinement were performed with isotropic B-factors, with incremental increases in resolution up to the resolution limit. Refinement was interleaved with manual model building using Coot[64] - with ligands, additional conformations of side chains and waters added after each stage of refinement. For all structures, occupancies of single conformation residues and ligands were set to one and adjusted manually based on the electron density. The occupancies of residues with double conformations were set to 50% and then refined in SHELXL, using the FVAR command. For the BrNiR dithionite-reduced structure, both T1Cu and T2Cu positions and the single solvent water were also automatically refined using the FVAR command in SHELXL. For AcNiR structures, only the occupancies of $Asp_{CAT}$ conformations were automatically refined in SHELXL using the FVAR command. Anisotropic B-factors were then refined, together with the addition of hydrogen atoms and at the final stage of the refinement, one cycle of unrestrained refinement (BLOC 1 style) was implemented to estimate the standard uncertainties (e.s.d.s) of coordinates and derived parameters (bond lengths and angles). Ellipsoidal models for all structures showing ADPs with 50% probability were also produced using the ShelXle GUI[65], with these shown in Supplementary Fig. 5. Data collection and refinement statistics are shown in Supplementary Table 2.

### Single crystal and solution UV-vis absorption spectroscopy
Single crystal UV-vis absorption spectroscopy measurements were obtained off-line at SPring-8 at 100 K using a fibre-optic micro-spectrophotometer with a linear charge-coupled device array detector (Ocean Optics, SD2000). Single crystals of $Br^{2D}$NiR, BrJNiR and AcNiR with dimensions ~400 $\mu m^3$ were used for measurements. Crystals were transferred to a storage solution as described above prior to crystal soaking. For BrNiR crystals, these were soaked in 2.8 M ammonium sulphate, 50 mM HEPES (pH 5.5) [+200 mM sodium nitrite or 100 mM sodium dithionite] before being soaked in the same solutions with 17.5 % sucrose for use as a cryoprotectant. and for high pH measurements, these were soaked similarly but with a buffer of 50 mM Tris pH 7.3 instead and for longer incubation times to allow for pH increase. For AcNiR crystals, these were soaked in 2.8 M ammonium sulphate, 50 mM citrate (pH 4.8) [+200 mM sodium nitrite or 100 mM sodium dithionite] before being soaked in the same solutions with 17.5 % sucrose for use as a cryoprotectant. For the experiment involving nitrite soaking of reduced crystals, these were performed on the chemically-reduced cryoprotected colourless samples, which were in the T1Cu$^{+}$ reduced state. Following reduction, crystals were transferred to a cryoprotectant solution containing 200 mM sodium nitrite. For short soaking, crystals were soaked ~ 20 seconds before being cryocooled by plunging into liquid nitrogen, which would cryotrap the state. For long soaking, crystals were soaked ~ 3 minutes before being cryocooled by plunging into liquid nitrogen, which would cryotrap the state. Solution UV-vis absorption spectroscopy measurements were obtained using a Cary 3500 Compact UV-Vis Spectrophotometer (Agilent) with a quartz cuvette of 10 mm pathlength. A final protein concentration and buffer of 26 μM in 50 mM HEPES pH 6.5 was used for all measurements and a reference containing just 50 mM HEPES pH 6.5 buffer was taken prior to

measurements for subsequent background subtraction. For nitrite addition, a final concentration of 10 mM sodium nitrite ($NaNO_2$) was added to the protein solution. Likewise, to reduce the enzyme, a final concentration of 1 mM dithionite was added. 10 mM $NaNO_2$ was added to the reduced form of the enzyme for the additional experiment, with collection started immediately after this.

All spectra were processed using the icOS-toolbox app[66]. The average absorbance between 500 and 510 nm was subtracted from every spectrum for baseline correction. Corrected spectra were plotted using the Igor Pro software package (Wavemetrics).

### Reporting summary

Further information on research design is available in the Nature Portfolio Reporting Summary linked to this article.

## Data availability

The atomic coordinates have been deposited in the Protein Data Bank (PDB ID codes: 9ROS (atomic resolution (1.15 Å) XFEL structure of oxidised as-isolated *Br*NiR at pH 5.5), 9RO1 (atomic resolution (1.00 Å) XFEL structure of oxidised as-isolated *Br*NiR at pH 7.3), 9RNZ (atomic resolution (1.02 Å) XFEL structure of nitrite-bound *Br*NiR at pH. 5.5), 9ROC (atomic resolution (1.05 Å) XFEL structure of dithionite-reduced *Br*NiR at pH. 5.5), 9RLL (sub-atomic resolution (0.95 Å) XFEL structure of oxidised as-isolated *Ac*NiR at pH 4.8), and 9RN0 (sub-atomic resolution (0.95 Å) XFEL structure of nitrite-bound *Ac*NiR at pH 4.8).

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

## Acknowledgements

This work was supported by the Biotechnology and Biological Sciences Research Council, UK (grant number BB/N013972/1 to S.S.H./S.V.A.). The XFEL experiments were performed at BL2 of SACLA with the approval of the Japan Synchrotron Radiation Research Institute (JASRI) (Proposal nos. 2021A804 and 2022A806). Samuel Rose was supported by the RIKEN–Liverpool Partnership, awarded to Masaki Yamamoto and Samar Hasnain. It was also partly supported by the Platform Project for Supporting Drug Discovery and Life Science Research (Basis for Supporting Innovative Drug Discovery and Life Science Research; BINDS) from the Japan Agency for Medical Research and Development (AMED) under Grant No. JP20am0101070. We thank Carlos Brondino and Tetsuya Ishikawa for their support and interest in the project. We would also like to thank Drs. Kanji Shimba, Machika Kataoka and Kotone Ishihara for their help.

## Author contributions

S.V.A., R.R.E., M.Y., and S.S.H. conceived and designed the project. S.L.R. expressed and purified Br²ᴰNiR and AcNiR, respectively, and F.M.F. expressed and purified BrJNiR. S.L.R. and S.V.A. crystallised all the proteins used here for XFEL and single-crystal spectroscopy experiments. S.L.R., S.V.A., H.S., T.T. and S.S.H. collected the SF-ROX and single crystal spectroscopy data. S.L.R. collected solution spectroscopy data. K.H., H.A., G.U. and H.M. provided support for data collection by implementing the automatic collection pipeline. K.Y. performed the SF-ROX data processing; S.L.R. and S.V.A. performed the SF-ROX structure determination and SHELXL refinement; S.L.R., R.R.E., and S.S.H. wrote the manuscript with contributions from all the authors. All authors have

given approval to the final version of the manuscript. M.Y., S.V.A. and S.S.H. provided the funding for the programme.

## Competing interests

The authors declare no competing interests.
