## [Transparent Peer Review file · Nature Communications]

True atomic resolution XFEL structures of a metalloenzyme reveal key insights into its catalytic mechanism

Corresponding Author: Professor Samar Hasnain

Version 0:

Reviewer comments:

Reviewer #1

(Remarks to the Author)

The manuscript by S. Samar Hasnain and colleagues is one of the first comprehensive studies of copper nitrite reductases, examining different oxidation states, and substrate binding at different pH values using X-ray energy. The atomic-resolution structures presented here provide valuable insights into the substrate-binding mode and the coordination of the two copper sites, which are of great importance to researchers working with nitrite reductases and other copper metalloproteins. The ability to attain information on resting and substrate-bound states also makes this study relevant to those working with enzymes.

The manuscript presents a substantial amount of new data and is generally well-written and clearly presented. The number and quality of figures are appropriate for the scope of the study.

However, there are several issues that require clarification or correction before the manuscript can be considered for publication.

Note: As the manuscript lacks page numbers, all comments below will be referenced to line numbers.

1. The abstract should be revised to present the main conclusions obtained in this study.

2. Line 50: Role in Nitrification

The authors state that the enzyme is a key enzyme in the nitrification pathway. To the best of my knowledge, this enzyme is primarily involved in the denitrification pathway. I would appreciate a clarification and appropriate referencing on how the enzyme fits into nitrification, or a correction if this was a misstatement.

Reference 3 is not appropriate to support this sentence. Please revise your references throughout the manuscript.

3. Lines 56 to 70: Missing references

This section includes several assertions and background details that should be supported by citations. Please provide appropriate references to substantiate the statements made here.

4. Classes of Nitrite Reductases – Clarify Earlier

The different classes of nitrite reductases used in this study are only introduced later in the manuscript (Line 395). For clarity and better flow, I strongly suggest that the authors introduce this classification earlier, ideally in the Introduction or at the beginning of the Results section. This would provide readers with the necessary context to interpret the visible spectra in Figure 3, that is first referenced in Line 221.

It would also be important to mention why these classes were chosen.

5. Methods: Missing information

The Methods section lacks essential details about the protein samples used for crystallization. In particular, the authors should provide more information on the metal content of the proteins, including the number of copper atoms bound per polypeptide chain. Additionally, data on protein purity, such as absorbance ratios, and the specific activity of the enzyme should be reported to confirm that the crystallized protein was functionally competent. The occupancy of the copper site obtained from structural refinement should also be included in the manuscript text, as this information is important to support the structural interpretation of the active site.

6. Line 155: Bound hydroxide at low pH

The manuscript reports a bound hydroxide ion at pH 4.8, yet it does not offer an explanation for the presence of OH⁻ at such an acidic pH. This observation is unusual and warrants further discussion, particularly with regard to the possible involvement of a nearby residue that might change the local pK_a, or to stabilization through metal coordination. Can the authors mention the occupancy to identify it was an oxygen atom.

7. Line 188: Coordination of T1Cu BrNiR

In the discussion of the T1Cu center in BrNiR, the authors mention that Met145 adopts two positions in the structure solved

at pH 7.3 and in the nitrite-bound form. However, no mechanistic or structural rationale is provided for this dual conformation. Although they cite literature (line 189-190) linking the axial methionine's position to the reduction potential of the T1Cu sites, they do not directly relate this to the properties of their own proteins.

Furthermore, such difference in geometry of the center should be observed in its spectroscopic features, such as visible absorption spectra, visible CD or EPR spectra. The visible spectra obtained from the crystals should be better analyzed to corroborate this observation. Moreover, the authors should mention whether this effect has been observed in the structures of other T1Cu proteins.

Can the authors explain whether such conformations cannot be attributed to low copper occupancy?

8. Line 425-427: Catalytic Mechanism

The reviewer can agree that, in the mechanism of these enzymes, nitrite binds before the electron is delivered, so that it binds to the oxidized T2Cu center (with or without T1Cu center reduced). However, the reviewer disagrees with the authors' explanation and interpretation of the spectroscopic data. The authors claim that the low recovery of the T1Cu absorption features means that nitrite has to bind before T2Cu is reduced. However, the recovery of the T1Cu absorption features depends on the relative amounts of the reducing agent and nitrite, as well as on the catalytic rate constant. The time at which the spectra were collected after the addition of nitrite is also a factor.

Therefore, to support their hypothesis of "nitrite binding-before-reduction", the authors could perform the reaction in reverse order (add nitrite to oxidized NiR and then dithionite) and collect spectra at the same time points as before.

9. pKa of the bell-shaped curve

The authors study the structure of what they designate as "high" and "low" pH. Most low pH falls below or are the optimum pH of these enzymes, with high pH above it. It is important to note the pKas of the pH dependence of the catalytic activity of each enzyme is not reported and it would help the interpretation of the data. Additionally, the authors fail to establish a clear correlation between the protonation state of AspCAT and HisCAT during catalysis. From their data AspCAT only becomes deprotonated after nitrite binding and at high pH, whereas HisCAT is always protonated as required for activity. Their structures would correspond to the optimum pH structures, but further data on proton transfer is not made.

10. Low pH structures

At this pH, is the protein damaged? Is the stability of these enzymes high at such a low pH?

Line 497: Are these low pH values the optimum pH of the enzymes under study? Please provide a reference.

11. Use of HEPES | MES

The authors mention several times in the Methods section to have used HEPES pH 5.0 or 5.5, this is a typo for sure, as the pKa of HEPES is around 7.5 and at such low pH it does not have buffering capacity. So, in which conditions HEPES was used?

Figures and Figure Legends

In the Figure legends add for all cases the indication of which pH the protein was crystallized (including Figure 5).

1. Line 200: Legend of Figure 2

Mention in the Figure legend the meaning of BW. As far as I can see GK is not in this figure. The figure legends need to be revised.

2. Line 299: Legend of Figure 4

Panel (b) seems to still show the two conformations in cyan and blue. Was this done on purpose?

3. Line 390: Legend of Figure 6

The Figure legend mentions LD, which is not in the figure. Please indicate the meaning of water WG and GK.

Other corrections

1. Page 46, The c as in cytochrome c should be in italic (please check the text for other occurrences).

2. Page 53, cd₁, 1 is subscript and not in italic

3. Line 353, indicate at which pH the protein was crystallized.

4. Line 401: japonicum without capital letter

5. Line 438-440: sentence needs to be revised

6. Line 453-456: "... allowed us to conclude that these residues ..." residue or residues, seems to be related only to HisCAT.

7. Line 485 (and other below): penta-coordinate substitute for penta-coordinated

8. Line 485: revise "tris-his coordination and two T2Cu ligands" – not usual designation and missing words?

9. Line 486: and a hydroxide

10. Line 558 (and other): microL or mg/mL (correct the Liter as capital letter)

11. Line 675: concentration and buffer

The designation microm, is a typo for microM? The buffer used was really HEPES pH 6.5? It would be on the limit of the buffering capacity. The spectra were collected after nitrite addition to the reduced protein at which time points? This should be mentioned in the Figure legend. The same should be mentioned for the oxidized form incubated with nitrite. Please check my comment above.

12. Line 686: to be removed?

13. References

The reference style needs to be revised as it is not homogenous throughout the document: different fonts used, different formats, missing DOI for most, name of the journals not uniform, bold year or volume, date sometimes in parenthesis, titles with capitalized words in some cases and in other not, microorganism's name should always be in italic.

Reviewer #2

(Remarks to the Author)

This manuscript presents, to the best of our knowledge, the first atomic resolution XFEL structures of copper nitrite reductases (CuNiRs) in multiple functional relevant states, as oxidized, reduced, and nitrite-bound forms. By exploiting high-

energy (13 keV) femtosecond X-ray pulses at SACLA and refining with SHELXL, the authors achieve structures free from radiation-induced chemistry, overcoming a major limitation of synchrotron studies where photoreduction distorts the copper sites. The work integrates crystallography with single-crystal spectroscopy to confirm the redox state of the type 1 copper site and supports new mechanistic insights into nitrite binding modes, water/hydroxide coordination, and the protonation states of key catalytic residues, such as Asp_{CAT} and His_{CAT}. These findings may have broad implications for understanding proton-coupled electron transfer, substrate gating and binding and reactivity in CuNiRs. Moreover this manuscript represents the first application of SHELX, a suite of crystallographic software, to XFEL atomic resolutions structures. The manuscript is well-written and the methods described in the main text, and the details in the supporting information, are enough detailed to allow reproduction. Overall, the work is exciting and deserve to be published in Nature Communications journal after clarification of the following points:

- 1) Figure 4 d, e, f are almost certainly type 1 not type 2 Cu sites as they contain cys and met residues. I find it hard to believe that if the coordination number of the copper changes (losing a water) and the copper shifts positions that the imidazoles of the ligands also do not rotate. Figure 4b implies they don't. If the authors want to really compare these structures they need to show conformations of all atoms at the active site. Figure 4d seems to account for these differences.
- 2) Page 18, they are inferring protonation states from bond lengths and angles not determining protonation states. The latter case, actual seeing the protons would require a neutron structure.
- 3) We get the point that these are "the first" structures. The authors should only state this once and then focus on what their results mean. The reader feels beaten over the head with this emphasis on novelty when this statement is constantly repeated in abstract, introduction, results and discussion.
- 4) The abbreviation for the bridging water is sometimes inconsistent. On page 7 it is written as W_{Br} (with Br as subscript), while in Figures 2a, 2b, and 2c it is labelled as BW. The notation should be standardized along the text and figures.
- 5) As inferred by the authors in the Figure 2 captions, the loop connecting the His ligands of the T1Cu and T2Cu sites is proposed to act as a "sensor" in nitrite binding. Given the atomic resolution of these structures, it would be nice to know whether the authors compared the conformation of this sensor loop across the different functional states (oxidized, reduced, nitrite-bound). For example, a analysis of the RMSD calculated on the Calfa atoms of this loop could highlight small relevant rearrangements. Including such a comparison may strengthen the argument for its role in substrate binding.
- 6) How the authors determined the occupancies of residues around the T1Cu and T2Cu centers? It is not entirely clear from the methods. Were the occupancies adjusted manually based on the electron density, or were restraints applied to refine? Given the importance of correctly modeling partial occupancies near the active site, a more detailed explanation in the Methods would be very helpful. The inclusion of specific refinement criteria to justify alternate conformations and their percentage would increase the strength of the results. In addition, including Fo-Fc omit maps in the supporting information could further support the presence of alternate conformations or partial occupancies.
- 7) In the BrCuNiR structure at pH 5.5 (Figure 2a), His250 appears to be involved in a different hydrogen-bonding network compared to the pH 7.3 structure (Figure 2b). Could the authors clarify whether they observe any difference in the orientation of His250 between these two conditions? Additionally, what is the role of the bridging water at pH 7.3? Does it form new hydrogen bonds respect the structure at pH 5.5? These points should be addressed, as subtle structural differences may provide a structural rationale for the lower catalytic activity observed at neutral pH.
- 8) The authors state that all structures were collected at 100 K. However, at cryogenic temperatures protein flexibility, hydrogen-bonding interactions etc. can differ significantly from those occurring in solution under physiological conditions. A short discussion of how cooling may influence the observed hydrogen-bond network, nitrite binding, and mechanistic interpretations would need to be added.
- 9) The primary concern with the paper is not in the results, which are highly significant, but in the lack of broader analysis in the discussion. Now that we have all these novel structures, what do they tell us about the catalytic mechanism of the system? We are given exhaustive detail on structures of both type 1 and 2 Cu centers, on the substrate bound forms and on various pH conditions. But in my opinion, none of this data is brought together to assess catalytic mechanism for nitrite reduction that have been proposed. For example, is proton gating critical and how is it achieved both by changing the catalytic site and also the electron transfer site? Are structural changes observed consistent with known reduction potential changes in these systems based on substrate binding and protonation? How is the nitrite conformation important for proposed mechanisms (they hint at this in Fig 2 results showing gatekeeper conformations but never discuss how this influences the type 1 center structure and hence potential)? I could go on... The authors have really addressed excellently critical structural issues with this work; however, they have dropped the ball when tying the importance of their discoveries to the known catalytic system. This discussion is what is required to elevate this article to the level of Nat Comm, otherwise, the work should be in a more specialized, but still high impact crystallographic journal. With the proper discussion, I would fully support Nat Comm publication.

Minor issues:

- Typos along the text like "SHLEXL" in the abstract
- Mixing british and american English spellings (e.g., oxidised vs oxidized, modelled vs modeled). Please standardize to one style, in line with the journal guidelines.

Reviewer #3

(Remarks to the Author)

Reviewer #4

(Remarks to the Author)

The authors report several atomic resolution crystal structures of copper nitrite reductase (CuNiR) from *Bradyrhizobium* and *Achromobacter* species obtained using an X-ray free electron laser (XFEL), providing detailed and radiation damage-free views of the two metal centers in these redox proteins. The authors used an approach where they rotate and serially expose multiple regions of larger crystals to the XFEL beam at cryogenic temperature (SF-ROX), which permits using many fewer crystals to obtain a complete dataset. SF-ROX has the clearly demonstrated potential to generate data whose resolution exceeds that typically obtained using more conventional serial delivery of randomly orientated microcrystals typically used at XFELs. The authors have previously published pioneering work on this method and applied it to CuNiR enzymes (PMID: 29354268, 31316819). Therefore, this is an established method and the principal advance being reported in this manuscript is obtaining markedly improved resolution data for CuNiRs from multiple species. This improved resolution allows them to refine the models in SHELXL, which can produce estimated standard uncertainties (ESUs) on model parameters. Overall, this is an impressive example of crystallography at XFEL sources being used to answer questions that are not easily accessible using other X-ray sources owing to radiation damage.

Despite its technical strengths, the central findings of the study are difficult to summarize, as they cover the details of the changes in the metal and coordinating residue geometries as a function of species, pH, and redox state. In support of this impression, the Abstract provides unusually little sense of what was learned in the work. While the results are discussed in exhaustive detail, it is unclear (at least to this reviewer) what the core general message of the manuscript is. For example, I cannot confidently determine what a reader (particularly one who is not actively working on the structural biology of CuNiRs) might take from this study. This is at least partially the result of a difficult writing style, focused on providing lists of interatomic distances in multiple proteins in text form, some of which appear to be of minor importance. Furthermore, the figures are not always clear, with key distances or interactions discussed in the manuscript being sometimes hard to locate in the corresponding figure. Overall, this technically impressive work lacks the clearly stated core message and easily identified conceptual advance that I would expect from a manuscript submitted to a general readership journal. I provide detailed comments below.

Major points:

1. As stated above, the lengthy discussion of structural details in the Results reads as a text summary of the PDB files, with little sense of the significance of many of the listed distances or contacts. However, this is not uniformly true, as the discussion of redox-dependent changes in the T1Cu and T2Cu centers is related back to mechanistic details in an interesting way. I wish the rest of the structural discussion was also firmly rooted in mechanistic aspects that are clearly explained to the reader. For example, the mechanism of the enzyme is never illustrated. In my opinion, this should be a core figure in the main text, possibly figure 1. The authors state on lines 116-117 that "These observations collectively indicate that the binding-before-reduction branch of the proposed "random sequential mechanism" predominates during turnover.", but this mechanism and the associated open question is never shown or further clarified. This is just one suggestion to address the general sense that manuscript describes a number of impressively high-resolution structures at a level of uncontextualized detail that is difficult for readers to follow.
2. The authors emphasize in several places the special importance of using SHELXL for refinement. Their point about using unrestrained refinement and estimated standard uncertainties (ESUs) to determine the protonation states of carboxylic acids (starting on line 430) is a nice illustration of a unique capability of SHELX, but it is not fully explored. Unrestrained refinement presumably provided ESUs on all distances, so why not use the parenthetical error notation used in this section to provide ESUs for all of the reported distances in the manuscript? As just one of many potential examples, on line 369, the authors state that the N-O distances are approximately equal (~ 1.27 Å). This would be a good place to use those ESUs to directly show that these bonds are equal to within 1-2 ESUs. Related to my suggestions in point 1 (above), there are several opportunities to more thoroughly analyze and better integrate the ESUs into the core scientific argument of the manuscript.
3. Related to point 1 above, a concluding summary figure that ties together the large amount of information in this manuscript would be very helpful for the reader. Points to address include: the important lessons learned by comparing reduced vs. oxidized structures, the key differences or similarities between the three species of enzyme studied, and the mechanistic implications of this study.
4. The authors might consider commenting of the potential influence of 100K data collection used here, which has been shown in a large and growing body of work to sometimes distort functionally relevant disorder/dynamics. Because one of the primary advantages of XFEL crystallography is the absence of radiation damage in room-temperature datasets, this point might be particularly important to mention.
5. The authors might also consider a deeper analysis of the anisotropic atomic displacement parameters (ADPs) that they were able to refine for these high resolution structures. For example, do preferred directions of atomic motion in the active site differ between these enzymes from different species, in different redox states, or at different pHs? Another possibility is to investigate whether the ADPs relate to observed disorder for the bound NO₂⁻, or if there is there any indication of how the top-hat vs. side-on conformations of NO₂⁻ differ in atomic displacements.

Minor points.

6. SHELXL is misspelled in multiple places and there are other typos that should be corrected.

7. Referencing to figures in the text would benefit from specifying panel, not just figure number.
8. Owing to the number of different proteins and states characterized, references to which protein and state are being discussed should be made clearer throughout (e.g. lines 222-227).
9. Abbreviations are not always defined, e.g. ET is never clearly defined as "electron transfer".
10. Some of the water molecules near the T2Cu active site are reported to be very close (e.g. 2.16 Å on line 216). The authors speculate that perhaps one is an OH⁻ anion. Would that permit these close distances? They still seem unexpectedly near to each other and it would be valuable to know the authors' thoughts on this. Were they identified as clashes during validation?
11. As above, the discussion of the spectroscopic data would be better integrated into the manuscript if mechanistic context, preferably in the form of one or more figures, were provided. As it is, these data seem more supplemental than central to the manuscript.

Reviewer #5

(Remarks to the Author)

Version 1:

Reviewer comments:

Reviewer #1

(Remarks to the Author)

The authors have thoroughly revised the manuscript attending to all the suggestions of the reviewers. The manuscript has been improved considerably, specially by the inclusion of final scheme. Some minor considerations:

1. Though not pointed out before, panel 1c is not really a figure and the reviewer suggests moving it to supplementary materials.
2. Figure 3: pH at which the experiment was performed should be stated in the figure legend (for the crystals and solution experiments).
3. Relative to this experiment, this reviewer does not agree with the statement that low recovery of absorbance when nitrite is added to the reduced enzyme, means that it is because nitrite does not bind to the reduced catalytic center. In the literature, similar experiments performed with other CuNir from other organism clearly show a complete or nearly complete recovery of the T1Cu absorption bands after 10 min. Either these enzymes have low catalytic rate constant or low T2Cu center occupancy upon reduction (might be explained by the values in table S3). See publications: 10.1039/d0mt00177e and 10.1016/j.bbagen.2017.10.011.
Another explanation could be the pH at which the experiment was performed, as at higher pH the activity of the enzyme decreases (buffer pH change with temperature). Can the authors comment on this.
This reviewer just does not agree with sentence in line 129-130 | 495-496 – "one branch of the random sequential mechanism predominates relative to the other", taken from the experiment shown in Fig.3.
4. Some typos are still found in the manuscript but will for sure be corrected at a later stage of the manuscript.

Reviewer #2

(Remarks to the Author)

The authors have addressed all my comments satisfactorily, and I now consider the manuscript suitable for publication in Nat.Comm.

Reviewer #3

(Remarks to the Author)

Reviewer #4

(Remarks to the Author)

The authors have addressed my concerns from the initial review. This manuscript is much-improved and clarifies key points that were missing before.

Reviewer #5

(Remarks to the Author)

Version 2:

Reviewer comments:

Reviewer #1

(Remarks to the Author)

The authors have thoroughly revised the manuscript attending to all the suggestions of the reviewers. The reviewer has no further comments.

Responses to reviewers comments

REVIEWER COMMENTS

Reviewer #1 (Remarks to the Author):

The manuscript by S. Samar Hasnain and colleagues is one of the first comprehensive studies of copper nitrite reductases, examining different oxidation states, and substrate binding at different pH values using X-ray energy. The atomic-resolution structures presented here provide valuable insights into the substrate-binding mode and the coordination of the two copper sites, which are of great importance to researchers working with nitrite reductases and other copper metalloproteins. The ability to attain information on resting and substrate-bound states also makes this study relevant to those working with enzymes. **The manuscript presents a substantial amount of new data and is generally well-written and clearly presented. The number and quality of figures are appropriate for the scope of the study.**

However, there are several issues that require clarification or correction before the manuscript can be considered for publication.

Note: As the manuscript lacks page numbers, all comments below will be referenced to line numbers.

Response: Revised manuscript has been page and line numbered.

1. The abstract should be revised to present the main conclusions obtain in this study.

Response: We have removed the generalised sentence "These structures, free from radiation induced chemistry with SHELXL refinement, have provided unprecedented insight into redox enzymes mechanisms." by "A consistent picture has emerged with the observation of a penta-coordinated catalytic type 2 Cu centre in the resting state in all cases. A tetra-coordinate reduced Cu site with single water has been observed for the first time giving structural support to the random-sequential scheme."

2. Line 50: Role in Nitrification: The authors state that the enzyme is a key enzyme in the nitrification pathway. To the best of my knowledge, this enzyme is primarily involved in the denitrification pathway. I would appreciate a clarification and appropriate referencing on how the enzyme fits into nitrification, or a correction if this was a misstatement.

Response: Though the bulk of research has focussed on its involvement in denitrification and nitrifier denitrification pathways. However, in recent years a CuNiR has been characterized from an anammox organism (energy generation involving the anaerobic conversion of ammonium to N₂ via nitrite). In the context of nitrification, the discovery that NO is produced by hydroxylamine reductase as an obligate intermediate in the first stage of nitrification, a role for NirK in catalysing the oxidation of NO to form nitrite has been proposed (JD Caranto & KM Lancaster [pnas.org/cgi/doi/10.1073/pnas.1704504114](https://doi.org/10.1073/pnas.1704504114)). Consistent with this, *nirK* is present in all published genomes of ammonia-oxidizing archaeobacteria including those which do not denitrify, strongly suggesting that it participates in multiple several steps of the nitrogen cycle.

Reference 3 is not appropriate to support this sentence. Please revise your references throughout the manuscript.

Response: This was an error. It has been replaced by references 6-8 including our recent review.

3. Lines 56 to 70: Missing references. This section includes several assertions and background details that should be supported by citations. Please provide appropriate references to substantiate the statements made here.

Response: We had mentioned references 6-12 on line 56. We have now broken it down and provided additional references going up to 17.

4. Classes of Nitrite Reductases –The different classes of nitrite reductases used in this study are only introduced later in the manuscript (Line 395). For clarity and better flow, I strongly suggest

that the authors introduce this classification earlier, ideally in the Introduction or at the beginning of the Results section.

Response: We did introduce different classes very early on in the Introduction. On Lines 52-54, we said “Two distinct classes of nitrite reductases are found in denitrifying organisms, a cytochrome *cd₁* type, encoded by *nirS* and the more widely distributed copper-containing enzymes encoded by *nirK*^{6,7}” and in the context of CuNiRs sub-classes, we mention blue and green CuNiRs on Line 58. We have added it also at line 177.

5. Methods: The Methods section lacks essential details about the protein samples used for crystallization. In particular, the authors should provide more information on the metal content of the proteins, including the number of copper atoms bound per polypeptide chain. Additionally, data on protein purity, such as absorbance ratios, and the specific activity of the enzyme should be reported to confirm that the crystallized protein was functionally competent. The occupancy of the copper site obtained from structural refinement should also be included in the manuscript text, as this information is important to support the structural interpretation of the active site.

Response: We added references to relevant papers, T2Cu occupancy we added to table S3. *BrjNiR* was purified as previously reported [Metallomics, 12(2020):2084–2097; Archives of Biochemistry and Biophysics, 770 (2025):110467]. The Cu content was determined using either the Biquinoline method or atomic absorption spectrometry. A ratio of 2 mol of Cu to 1 mol of *BrjNiR* monomer is usually obtained after SEC, with a minimum of 1.9. Another checkpoint is the A_{280}/A_{592} ratio, but this depends on the oxidation state of the T1Cu centre. Good, *as-purified* samples usually yield a ratio of 16–18. To check the integrity of the T1Cu and T2Cu centres, the samples were subjected to EPR analysis. Activity assays were performed using MV, yielding a value $\sim 15 \mu\text{mol nitrite min}^{-1} \text{mg}^{-1}$ monomeric *BrjNiR*. Specific activity of *AcNiR* and *BrNiR* has been reported in (Daisuke et al), *AcNiR/BrNiR* activity is 924/41 $\text{nmol s}^{-1}(\text{nmol of protein})^{-1}$, respectively. The samples used for crystallisation have similar activity. Both samples were fully Cu loaded.

6. Line 155: Bound hydroxide at low pH. The manuscript reports a bound hydroxide ion at pH 4.8, yet it does not offer an explanation for the presence of OH^- at such an acidic pH. This observation is unusual and warrants further discussion, particularly with regard to the possible involvement of a nearby residue that might change the local pKa, or to stabilization through metal coordination. Can the authors mention the occupancy to identify it was an oxygen atom.

Response: We do discuss this and say that “the proximal position of the catalytic Aspartic residue (*Asp_{CAT}*) is also present, hydrogen bonding to W1 ligand”. We have rephrased it to clarify further.

7. Line 188: Coordination of T1Cu *BrNiR*. In the discussion of the T1Cu center in *BrNiR*, the authors mention that Met145 adopts two positions in the structure solved at pH 7.3 and in the nitrite-bound form. However, no mechanistic or structural rationale is provided for this dual conformation. Although they cite literature (line 189-190) linking the axial methionine's position to the reduction potential of the T1Cu sites, they do not directly relate this to the properties of their own proteins. Furthermore, such difference in geometry of the center should be observed in its spectroscopic features, such as visible absorption spectra, visible CD or EPR spectra. The visible spectra obtained from the crystals should be better analyzed to corroborate this observation. Moreover, the authors should mention whether this effect has been observed in the structures of other T1Cu proteins.

Response: We had avoided commenting on this in view of partial occupancy but now have included optical spectra for both *Rhizobia* CuNiRs at high pH also. Only in *Br^{2D}CuNiR*, which shows dual conformation of Met145, optical band at 590nm shows a significant reduction in intensity. For *BrjNiR*, the optical spectra at two pH are indistinguishable. We have included this in the text.

8. Line 425-427: Catalytic Mechanism. The reviewer can agree that, in the mechanism of these enzymes, nitrite binds before the electron is delivered, so that it binds to the oxidized T2Cu center (with or without T1Cu center reduced). However, the reviewer disagrees with the authors' explanation and interpretation of the spectroscopic data. The authors claim that the low recovery of the T1Cu absorption features means that nitrite has to bind before T2Cu is reduced. However, the recovery of the T1Cu absorption features depends on the relative amounts of the reducing agent and nitrite, as well as on the catalytic rate constant. The time at which the spectra were collected after the addition of nitrite is also a factor. Therefore, to support their hypothesis of “nitrite

binding-before-reduction", the authors could perform the reaction in reverse order (add nitrite to oxidized NiR and then dithionite) and collect spectra at the same time points as before.

Response: We appreciate reviewer's suggestion of performing the reaction in reverse order. However, we have shown (PNAS 2022) in crystal through MSOX studies of oxidised and nitrite bound oxidised enzyme that electron transfer from T1Cu to T2Cu occurs only when nitrite is bound to T2Cu. The same has been shown by us in laser-flash photolysis and optical spectroscopy experiments which showed rapid ET from photoexcited NADH to the T1Cu centre but little or no inter-Cu ET in the absence of nitrite.

9. pKa of the bell-shaped curve. The authors study the structure of what they designate as "high" and "low" pH. Most low pH falls below or are the optimum pH of these enzymes, with high pH above it. It is important to note the pKas of the pH dependence of the catalytic activity of each enzyme is not reported and it would help the interpretation of the data. Additionally, the authors fail to establish a clear correlation between the protonation state of AspCAT and HisCAT during catalysis. From their data AspCAT only becomes deprotonated after nitrite binding and at high pH, whereas HisCAT is always protonated as required for activity. Their structures would correspond to the optimum pH structures, but further data on proton transfer is not made. Change low pH to optimum activity pH

Response: We have replaced low and high pH from the sub-headings and given the actual pH values for the structures.

10. Low pH structures. At this pH, is the protein damaged? Is the stability of these enzymes high at such a low pH? Line 497: Are these low pH values the optimum pH of the enzymes under study?

Response: CuNiRs are located in the periplasm and are generally robust with regard to temperature and extremes of pH covered by the pH-range here. For AxNiR the decrease in activity at pH values below the optimum was not due to instability or denaturation of the enzyme since all reactions were linear during the assays down to pH 4.2. (Abraham et al Biochem. J. (1997) 324, 511- 516). Lower pH values structures are at pH close to the optimum activity pH (pH6).

11. Use of HEPES | MES. The authors mention several times in the Methods section to have used HEPES pH 5.0 or 5.5, this is a typo for sure, as the pKa of HEPES is around 7.5 and at such low pH it does not have buffering capacity. So, in which conditions HEPES was used?

Response: There is no mistake. HEPES solution for crystallisation and soaking of BrNiR crystals was acidified by HCl from 6.8 to a pH of 5.5. As is always the case, buffered protein was centrifuged prior to crystallisation.

Figures and Figure Legends. In the Figure legends add for all cases the indication of which pH the protein was crystallized (including Figure 5).

Response: This has been done.

1. Line 200: Legend of Figure 2. Mention in the Figure legend the meaning of BW. As far as I can see GK is not in this figure. The figure legends need to be revised.

Response: GK and P conformations of Asp are clearly marked in panel f. Abbreviations have been explained.

2. Line 299: Legend of Figure 4. Panel (b) seems to still show the two conformations in cyan and blue. Was this done on purpose?

Response: Yes. The cyan conformation represents the fully reduced tri-coordinate Cu(I) site where flipping of the Ile252 residue is observed.

3. Line 390: Legend of Figure 6. The Figure legend mentions LD, which is not in the figure. Please indicate the meaning of water WG and GK.

Response: LD has been replaced by GK. Abbreviations has been explained.

Other corrections.

1. Page 46, The c as in cytochrome c should be in italic (please check the text for other occurrences). **DONE**

2. Page 53, cd₁, 1 is subscript and not in italic. **DONE**

3. Line 353, indicate at which pH the protein was crystallized. **DONE**
4. Line 401: japonicum without capital letter. **DONE**
5. Line 438-440: sentence needs to be revised. **DONE**
6. Line 453-456: "... allowed us to conclude that these residues ..." residue or residues, seems to be related only to HisCAT. **CORRECTED**
7. Line 485 (and other below): penta-coordinate substitute for penta-coordinated. **DONE**
8. Line 485: revise "tris-his coordination and two T2Cu ligands" – not usual designation and missing words?
Response: Changed to "T2Cu(II) to be penta-coordinate, with a (His)₃ coordination and two solvent ligands, either two waters or a water and a hydroxide."
9. Line 486: and a hydroxide (**see above point 8**)
10. Line 558 (and other): microL or mg/mL (correct the Liter as capital letter). **DONE**
11. Line 675: concentration and buffer. The designation microm, is a typo for microM? **DONE**
The buffer used was really HEPES pH 6.5? It would be on the limit of the buffering capacity. The spectra were collected after nitrite addition to the reduced protein at which time points? This should be mentioned in the Figure legend. The same should be mentioned for the oxidized form incubated with nitrite. Please check my comment above. We have addressed this under crystallisation section of the methods. **See response to point 11 above.**
12. Line 686: to be removed?
Done
13. References. The reference style needs to be revised as it is not homogenous throughout the document: different fonts used, different formats, missing DOI for most, name of the journals not uniform, bold year or volume, date sometimes in parenthesis, titles with capitalized words in some cases and in other not, microorganism's name should always be in italic. **DONE. We have not added DOIs.**

Reviewer #2 (Remarks to the Author):

This manuscript presents, to the best of our knowledge, the first atomic resolution XFEL structures of copper nitrite reductases (CuNiRs) in multiple functional relevant states, as oxidized, reduced, and nitrite-bound forms. By exploiting high-energy (13 keV) femtosecond X-ray pulses at SACLA and refining with SHELXL, the authors achieve structures free from radiation-induced chemistry, overcoming a major limitation of synchrotron studies where photoreduction distorts the copper sites. The work integrates crystallography with single-crystal spectroscopy to confirm the redox state of the type 1 copper site and supports new mechanistic insights into nitrite binding modes, water/hydroxide coordination, and the protonation states of key catalytic residues, such as Asp_{CAT} and His_{CAT}. These findings may have broad implications for understanding proton-coupled electron transfer, substrate gating and binding and reactivity in CuNiRs. Moreover this manuscript represents the first application of SHELX, a suite of crystallographic software, to XFEL atomic resolutions structures. The manuscript is well-written and the methods described in the main text, and the details in the supporting information, are enough detailed to allow reproduction. **Overall, the work is exciting and deserve to be published in Nature Communications journal after clarification of the following points:**

1) Figure 4 d, e, f are almost certainly type 1 not type 2 Cu sites as they contain cys and met residues. I find it hard to believe that if the coordination number of the copper changes (losing a water) and the copper shifts positions that the imidazoles of the ligands also do not rotate. Figure

4b implies they don't. If the authors want to really compare these structures they need to show conformations of all atoms at the active site. Figure 4d seems to account for these differences.

Response: We do clearly state that in Figure 4, panels d, e and f are for T1Cu site. Panels a to c are for T2Cu site where panel b and c show two conformations of reduced state (both deduced from the single structure of the chemically reduced T2Cu site shown in panel a). Panel a clearly shows that imidazoles are single conformation with no smearing of electron density,

2) Page 18, they are inferring protonation states from bond lengths and angles not determining protonation states. The latter case, actual seeing the protons would require a neutron structure.

Response: We agree but are reassured of our assignment of the protonation state, as we have shown in 2019 that the protonation state deduced from the high-resolution XFEL crystallography structures of AxNiR in the resting state agrees with neutron crystallography of perdeuterated enzyme (Reference 15 "Catalytically important damage-free structures of a copper nitrite reductase obtained by femtosecond X-ray laser and room-temperature neutron crystallography").

3) We get the point that these are "the first" structures. The authors should only state this once and then focus on what their results mean. The reader feels beaten over the head with this emphasis on novelty when this statement is constantly repeated in abstract, introduction, results and discussion.

Response: DONE

4) The abbreviation for the bridging water is sometimes inconsistent. On page 7 it is written as W_{Br} (with Br as subscript), while in Figures 2a, 2b, and 2c it is labelled as BW. The notation should be standardized along the text and figures.

Response: Corrected.

5) As inferred by the authors in the Figure 2 captions, the loop connecting the His ligands of the T1Cu and T2Cu sites is proposed to act as a "sensor" in nitrite binding. Given the atomic resolution of these structures, it would be nice to know whether the authors compared the conformation of this sensor loop across the different functional states (oxidized, reduced, nitrite-bound). For example, an analysis of the RMSD calculated on the Calfa atoms of this loop could highlight small relevant rearrangements. Including such a comparison may strengthen the argument for its role in substrate binding.

Response: We have done this comparison, but changes are insignificant to be included in the manuscript. We note that despite the atomic resolution, the quality of maps is not as good as what we have obtained at synchrotron storage ring at similar resolution. For example, in AcNiR at 0.87Å, synchrotron X-ray structure, we could identify ~1650 hydrogen atoms compared to ~2700 expected hydrogen atoms (see Figures 3 & 4 in *IUCrJ* (2015). 2, 464–474). There are potentially several reasons which need further systematic work to be undertaken, e.g. storage rings are DC stable sources while XFEL are pulsed sources where each pulse is a new source with significant variations and multiple crystals are used with each providing multiple spots. We prefer not to include such comments in the manuscript but are sharing with the reviewer.

6) How the authors determined the occupancies of residues around the T1Cu and T2Cu centers? It is not entirely clear from the methods. Were the occupancies adjusted manually based on the electron density, or were restraints applied to refine? Given the importance of correctly modeling partial occupancies near the active site, a more detailed explanation in the Methods would be very helpful. The inclusion of specific refinement criteria to justify alternate conformations and their percentage would increase the strength of the results. In addition, including Fo-Fc omit maps in the supporting information could further support the presence of alternate conformations or partial occupancies.

Response: Positions of residues with multiple conformations were modelled manually and assigned equal number, for example 50% for double conformations or 30% for side chains with 3 conformations and then refined by SHELX. Within T1Cu and T2Cu neighbourhoods, occupancies of the residues with a single conformation were set to 1.0; We included this in the methods section.

7) In the BrCuNiR structure at pH 5.5 (Figure 2a), His250 appears to be involved in a different hydrogen-bonding network compared to the pH 7.3 structure (Figure 2b). Could the authors clarify whether they observe any difference in the orientation of His250 between these two conditions?

Additionally, what is the role of the bridging water at pH 7.3? Does it form new hydrogen bonds respect the structure at pH 5.5? These points should be addressed, as subtle structural differences may provide a structural rationale for the lower catalytic activity observed at neutral pH. Response: At high pH His_{CAT} has stronger bond with W2 (2.9 Å), than with bridging water 3.4 Å, while at low pH the distances are almost equal. We have expanded discussion on changes in His_{CAT}. For example, in lines 167-171, we have written “The catalytic Histidine residue (His250; His_{CAT}) bridges Asp_{CAT} through a conserved bridging water (W_{Br}) and its imidazole ring is rotated towards Glu274, with its N^{δ1} atom hydrogen bonding with the carbonyl oxygen of Glu274 at 2.70 Å (Supplementary Fig. 3a). This is consistent with the previous SF-ROX structure at 1.3 Å resolution¹⁶.”

8) The authors state that all structures were collected at 100 K. However, at cryogenic temperatures protein flexibility, hydrogen-bonding interactions etc. can differ significantly from those occurring in solution under physiological conditions. A short discussion of how cooling may influence the observed hydrogen-bond network, nitrite binding, and mechanistic interpretations would need to be added.

Response: This is a very pertinent comment which we had not addressed. There is always a problem of correlating biophysical studies of enzyme catalysis and how they reflect the reality of the different physiological environments that enzymes are exposed to *in vivo*. In the case of CuNiRs this is reflected in differences between the extent of inter Cu ET in solution and *in crystallo*. Never-the-less our MSOX studies of nitrite-soaked crystals have provided unprecedented insight to the detail of structural changes that occur during the reduction of nitrite by CuNiRs in crystals at cryogenic temperatures. These MSOX studies have also been carried out at elevated temperatures (>190K), above the glass transition temperature,, permissive for conformational changes or H-bonding interactions to occur. Room temperature MSOX studies have also shown a consistent picture but of course with limited number of intermediates and rapid loss of resolution due to wider radiation damage. The ability of nitrite crystals to undergo the reduction of nitrite, formation of NO adducts and then its loss has been well documented in multiple MSOX studies (*IUCrJ* (2018). 5, 283–292, <https://doi.org/10.1016/j.jmb.2024.168706>, <https://doi.org/10.1073/pnas.2205664119>, etc). In the context of the comment raised by the referee, nitrite was added to the crystal at RT before data collection at 100 K. We have previously shown X-ray generated electrons in MSOX studies reduce the T1 centre first and that in the presence of nitrite, ET and catalysis occur. We consider that this makes the FRIC structures presented here directly relevant to single turnover events during catalysis. This has been clarified in the revised manuscript, and we thank the referee for raising this issue.

9) The primary concern with the paper is not in the results, which are highly significant, but in the lack of broader analysis in the discussion. Now that we have all these novel structures, what do they tell us about the catalytic mechanism of the system? We are given exhaustive detail on structures of both type 1 and 2 Cu centers, on the substrate bound forms and on various pH conditions. But in my opinion, none of this data is brought together to assess catalytic mechanism for nitrite reduction that have been proposed. For example, is proton gating critical and how is it achieved both by changing the catalytic site and also the electron transfer site? Are structural changes observed consistent with known reduction potential changes in these systems based on substrate binding and protonation? How is the nitrite conformation important for proposed mechanisms (they hint at this in Fig 2 results showing gatekeeper conformations but never discuss how this influences the type 1 center structure and hence potential)? I could go on... The authors have really addressed excellently critical structural issues with this work; however, they have dropped the ball when tying the importance of their discoveries to the known catalytic system. This discussion is what is required to elevate this article to the level of Nat Comm, otherwise, the work should be in a more specialized, but still high impact crystallographic journal. With the proper discussion, I would fully support Nat Comm publication.

Response: We have included a scheme outlining the main findings. Both PCET and Gating of substrate are well documented (references 22-25 and 28) and are indicated on the scheme. The scheme highlights a number of new findings including (1) resting state for all of the NiRs being penta-coordinate, (2) dithionite reduced despite the complete loss of T1Cu signal, being in two state, one where T2Cu has tetra-coordination with single water molecule that would be capable of binding the substrate and turn-over. The random-sequential mechanism was based on solution studies where dithionite treatment would have been assumed to reduce the T1Cu to fully reduced

state due to the loss of colour, (3) at high pH, the bridge between His_{CAT} and Asp_{CAT} is lost with His_{CAT} bridging to one of the T2Cu-coordinated solvent ligands providing a structural explanation for lower activity at higher pH, (4) at low pH, one of the coordinated waters is quite short suggesting a potential water – hydroxide (OH⁻) resting state of the enzymes, consistent with the proposed based on high-resolution neutron crystallography and computational studies^{53,54}.

Minor issues:

- Typos along the text like “SHLEXL” in the abstract. **DONE**

- Mixing british and american English spellings (e.g., oxidised vs oxidized, modelled vs modeled). Please standardize to one style, in line with the journal guidelines. **DONE**

Reviewer #3 (Remarks to the Author):

Reviewer #4 (Remarks to the Author):

The authors report several atomic resolution crystal structures of copper nitrite reductase (CuNiR) from Bradyrhizobium and Achromobacter species obtained using an X-ray free electron laser (XFEL), providing detailed and radiation damage-free views of the two metal centers in these redox proteins. The authors used an approach where they rotate and serially expose multiple regions of larger crystals to the XFEL beam at cryogenic temperature (SF-ROX), which permits using many fewer crystals to obtain a complete dataset. SF-ROX has the clearly demonstrated potential to generate data whose resolution exceeds that typically obtained using more conventional serial delivery of randomly orientated microcrystals typically used at XFELs. The authors have previously published pioneering work on this method and applied it to CuNiR enzymes (PMID: 29354268, 31316819). Therefore, this is an established method and the principal advance being reported in this manuscript is obtaining markedly improved resolution data for CuNiRs from multiple species. This improved resolution allows them to refine the models in SHELXL, which can produce estimated standard uncertainties (ESUs) on model parameters. **Overall, this is an impressive example of crystallography at XFEL sources being used to answer questions that are not easily accessible using other X-ray sources owing to radiation damage.**

Despite its technical strengths, the central findings of the study are difficult to summarize, as they cover the details of the changes in the metal and coordinating residue geometries as a function of species, pH, and redox state. In support of this impression, the Abstract provides unusually little sense of what was learned in the work.

Response: Thank you for your positive comments. We have summarised some of the key findings in the abstract and Scheme 1.

While the results are discussed in exhaustive detail, it is unclear (at least to this reviewer) what the core general message of the manuscript is. For example, I cannot confidently determine what a reader (particularly one who is not actively working on the structural biology of CuNiRs) might take from this study. This is at least partially the result of a difficult writing style, focused on providing lists of interatomic distances in multiple proteins in text form, some of which appear to be of minor importance. Furthermore, the figures are not always clear, with key distances or interactions discussed in the manuscript being sometimes hard to locate in the corresponding figure. **Overall, this technically impressive work lacks the clearly stated core message and easily identified conceptual advance that I would expect from a manuscript submitted to a general readership journal.**

Response: We thank the reviewer. It has encouraged us to revise the discussion and include a scheme summarising the new findings pertinent to enzyme mechanism is included.

I provide detailed comments below.

Major points:

1. As stated above, the lengthy discussion of structural details in the Results reads as a text summary of the PDB files, with little sense of the significance of many of the listed distances or contacts. However, this is not uniformly true, as the discussion of redox-dependent changes in the T1Cu and T2Cu centers is related back to mechanistic details in an interesting way. I wish the rest of the structural discussion was also firmly rooted in mechanistic aspects that are clearly explained to the reader. For example, the mechanism of the enzyme is never illustrated. In my opinion, this should be a core figure in the main text, possibly figure 1. The authors state on lines 116-117 that “These observations collectively indicate that the binding-before-reduction branch of the proposed “random sequential mechanism” predominates during turnover.”, but this mechanism and the associated open question is never shown or further clarified. This is just one suggestion to address the general sense that manuscript describes a number of impressively high-resolution structures at a level of uncontextualized detail that is difficult for readers to follow.

Response: We have revised the manuscript (Results and Discussion significantly) and included a scheme summarising the key findings.

2. The authors emphasize in several places the special importance of using SHELXL for refinement. Their point about using unrestrained refinement and estimated standard uncertainties (ESUs) to determine the protonation states of carboxylic acids (starting on line 430) is a nice illustration of a unique capability of SHELX, but it is not fully explored. Unrestrained refinement presumably provided ESUs on all distances, so why not use the parenthetical error notation used in this section to provide ESUs for all of the reported distances in the manuscript? As just one of many potential examples, on line 369, the authors state that the N-O distances are approximately equal ($\sim 1.27 \text{ \AA}$). This would be a good place to use those ESUs to directly show that these bonds are equal to within 1-2 ESUs. Related to my suggestions in point 1 (above), there are several opportunities to more thoroughly analyze and better integrate the ESUs into the core scientific argument of the manuscript. **Response: We have done this throughout the Results section and Discussion. We have also included Supplementary Tables 2 and 3 with these details.**

3. Related to point 1 above, a concluding summary figure that ties together the large amount of information in this manuscript would be very helpful for the reader. Points to address include: the important lessons learned by comparing reduced vs. oxidized structures, the key differences or similarities between the three species of enzyme studied, and the mechanistic implications of this study.

Response: We have included Scheme 1 as per suggestion. We are grateful to the reviewer for pointing this shortcoming.

4. The authors might consider commenting of the potential influence of 100K data collection used here, which has been shown in a large and growing body of work to sometimes distort functionally relevant disorder/dynamics. Because one of the primary advantages of XFEL crystallography is the absence of radiation damage in room-temperature datasets, this point might be particularly important to mention.

Response: We have added this in the concluding paragraph “We conclude that the ability to obtain atomic resolution damage free structures for a variety of functionally relevant states using few (50) crystals of $\sim 30 \mu\text{m}$ thicknesses by deploying 13keV X-rays from an XFEL in conjunction with the SF-ROX methods holds a promise for elucidating enzyme mechanism for many metalloenzymes. SF-ROX method, like SFX, is also applicable for ambient temperature crystallography but the propagation length for damage resulting from the first femtoseconds pulse is going to be significantly larger thus requiring a much larger number of crystals compared to those used here. The extent of propagation that may occur at room temperatures requires a systematic study.”

5. The authors might also consider a deeper analysis of the anisotropic atomic displacement parameters (ADPs) that they were able to refine for these high resolution structures. For example, do preferred directions of atomic motion in the active site differ between these enzymes from different species, in different redox states, or at different pHs? Another possibility is to investigate whether the ADPs relate to observed disorder for the bound NO_2^- , or if there is there any indication of how the top-hat vs. side-on conformations of NO_2^- differ in atomic displacements.

Response: We have included supplementary figure 4, showing ellipsoidal models of the T2Cu active sites in these structures.

Minor points.

6. SHELXL is misspelled in multiple places and there are other typos that should be corrected. **DONE**

7. Referencing to figures in the text would benefit from specifying panel, not just figure number. **DONE**

8. Owing to the number of different proteins and states characterized, references to which protein and state are being discussed should be made clearer throughout (e.g. lines 222-227). **DONE**

9. Abbreviations are not always defined, e.g. ET is never clearly defined as “electron transfer”. **DONE**

10. Some of the water molecules near the T2Cu active site are reported to be very close (e.g. 2.16 Å on line 216). The authors speculate that perhaps one is an OH⁻ anion. Would that permit these close distances? They still seem unexpectedly near to each other and it would be valuable to know the authors' thoughts on this. Were they identified as clashes during validation?

Response: We have included a Fo-Fc OMIT map (Supplementary Fig. 1b), which when compared to supplementary Figure 1(a) persuades us that both W1 and W2 exists. It is possible that at high pH W1 may be split into two but have left it as a single W1 to avoid over interpretation. W2 has only half occupancy. There were no clashes in validation report.

11. As above, the discussion of the spectroscopic data would be better integrated into the manuscript if mechanistic context, preferably in the form of one or more figures, were provided. As it is, these data seem more supplemental than central to the manuscript.

Response: The optical spectrum is incorporated in the presentation of the results throughout the manuscript (lines 173-176, lines 220-228, line 244, line 301, line 405, line 425 etc). We hope that the reviewer would be agreeable to keep it in the main text.

Reviewer #5 (Remarks to the Author):

Response to remaining points of Reviewer 1.

The authors have thoroughly revised the manuscript attending to all the suggestions of the reviewers. The manuscript has been improved considerably, specially by the inclusion of final scheme. **Some minor considerations:**

Response: We appreciate that reviewer is satisfied except for minor points which have responded in detail so that the Editor can be persuaded to accept the manuscript without further reiterations.

1. Though not pointed out before, panel 1c is not really a figure and the reviewer suggests moving it to supplementary materials.

Response: We have removed Figure 1C and included it in the supplementary material.

2. Figure 3: pH at which the experiment was performed should be stated in the figure legend (for the crystals and solution experiments).

Response: We have included this information.

3. Relative to this experiment, this reviewer does not agree with the statement that low recovery of absorbance when nitrite is added to the reduced enzyme, means that it is because nitrite does not bind to the reduced catalytic center. In the literature, similar experiments performed with other CuNir from other organism clearly show a complete or nearly complete recovery of the T1Cu absorption bands after 10 min. Either these enzymes have low catalytic rate constant or low T2Cu center occupancy upon reduction (might be explained by the values in table S3). See publications: 10.1039/d0mt00177e and 10.1016/j.bbagen.2017.10.011.

Response: We DO NOT say that nitrite does not bind to the reduced enzyme. We say reactivity with reduced enzyme is sluggish. Also, one of the authors (Ferroni) of the current paper is a lead or a corresponding author of the above papers which the reviewer cites. In addition, in 2012 Ferroni et al (<https://doi.org/10.1016/j.jinorgbio.2012.04.016>) had concluded from their experiments that either the nitrite binding is limited when T2Cu is reduced or that the T1Cu-T2Cu ET is affected. There is substantial literature that support the observation that reduced type 2 Cu site are unable to bind or weakly bind nitrite or inhibitory ligands such as azide, and to that reduce T2Cu site react very sluggishly with nitrite leading to only a slow re-oxidation of the type 1 centre (J. Mol. Biol. (1999) 287, 1001; J Biol Chem (1997), 272, 28455). In last of these papers (JBC) structure of nitrite bound to A_nNiR with low occupancy of nitrite. For further discussion on pH aspect see response below.

We have modified the text to say, “The low recovery of T1Cu optical spectrum (~20 %) indicates that binding of nitrite to the reduced T2Cu(I) site is limited but not prevented,....”

Another explanation could be the pH at which the experiment was performed, as at higher pH the activity of the enzyme decreases (buffer pH change with temperature). Can the authors comment on this.

Response: As suggested by the reviewer, this is a real issue for solution studies (see A Temperature Independent pH (TIP) Buffer for Biomedical Biophysical Applications at Low Temperatures (Chem Commun 2008, 7, 823–825. doi:10.1039/b714446f). We used Tris/HCL, HEPES and Citrate buffers, all of which will become significantly more acidic at cryotemperatures in solution.

In the case of flash-freezing of crystals of BrNiR, as in our current paper, we have shown in our multiple MSOX studies that enzyme turnover at pH 5.5 is not affected (*PNAS* 2022 *pnas.2205664119*, *j.jmb.2024.168706*, *Chem. Sci.*, 2020, 11, 12485–12492, etc). We have shown in optically validated MSOX studies that PCET from T1Cu to T2Cu is disrupted resulting in a significantly slower conversion of substrate (*j.jmb.2024.168706*). In these papers we have demonstrated that nitrite can be soaked to full occupancy and that metal sites have mostly full occupancy of Cu. Solution metal analysis in general are not specific to a site but to the overall metal content.

This reviewer just does not agree with sentence in line 129-130 | 495-496 – “one branch of

the random sequential mechanism predominates relative to the other", taken from the experiment shown in Fig.3.

Response: This is a position appears to be a conviction. Our response is simply against such a belief.

4. Some typos are still found in the manuscript but will for sure be corrected at a later stage of the manuscript.

Response: We have gone through the manuscript carefully again.